# QUANTIZED LOCAL INDEPENDENCE DISCOVERY FOR FINE-GRAINED CAUSAL DYNAMICS LEARNING IN REINFORCEMENT LEARNING

## ABSTRACT

Incorporating causal relationships between the variables into dynamics learning has emerged as a promising approach to enhance robustness and generalization in reinforcement learning (RL). Recent studies have focused on examining conditional independences and leveraging only relevant state and action variables for prediction. However, such approaches tend to overlook local independence relationships that hold under certain circumstances referred as event. In this work, we present a theoretically-grounded and practical approach to dynamics learning which discovers such meaningful events and infers fine-grained causal relationships. The key idea is to learn a discrete latent variable that represents the pair of event and causal relationships specific to the event via vector quantization. As a result, our method provides a fine-grained understanding of the dynamics by capturing event-specific causal relationships, leading to improved robustness and generalization in RL. Experimental results demonstrate that our method is more robust to unseen states and generalizes well to downstream tasks compared to prior approaches. In addition, we find that our method successfully identifies meaningful events and recovers event-specific causal relationships.

## 1 INTRODUCTION

Model-based reinforcement learning (MBRL) has showcased its capability of solving various sequential decision making problems (Kaiser et al., 2020; Schrittwieser et al., 2020). Since learning accurate and robust dynamics model is crucial in MBRL, recent works incorporate the causal relationships between the variables into dynamics learning (Wang et al., 2022; Ding et al., 2022). Unlike the traditional dense models that employ the whole state and action variables to predict the future state, causal dynamics models utilize only relevant variables by examining conditional independences. As a result, they are more robust to spurious correlations and generalize well to unseen states by discarding unnecessary dependencies.

Our motivation stems from the observation that the dependencies between the variables often exist only under certain circumstances in many practical scenarios. For instance, in the context of autonomous driving, a lane change is contingent on the absence of nearby cars within a specific distance range. Thus, it is crucial for autonomous vehicles to recognize and understand circumstances in which lane changes do or do not affect other vehicles. Our hypothesis is that the agent capable of reasoning such fine-grained causal relationships would generalize well to downstream tasks.

In this work, we aim to incorporate local independence relationship between the variables, which holds under certain contexts but does not hold in general (Boutilier et al., 2013), into dynamics modeling for improving robustness and generalization in MBRL. Unfortunately, prior causal dynamics models examining conditional independences are not capable of harnessing them. An alternative way is to estimate variables dependencies for each individual sample (Pitis et al., 2020; Hwang et al., 2023). However, such sample-specific approaches do not explicitly capture meaningful contexts that exhibit fine-grained causal relationships, making them prone to overfitting and less robust on unseen states.

**Contribution.** We present a new causal dynamics model that (i) decomposes the data domain into subgroups which we call events, (ii) discovers local independences under each event, and (iii) employs only locally relevant variables for prediction (Fig. 1). Clearly, it is crucial to discover

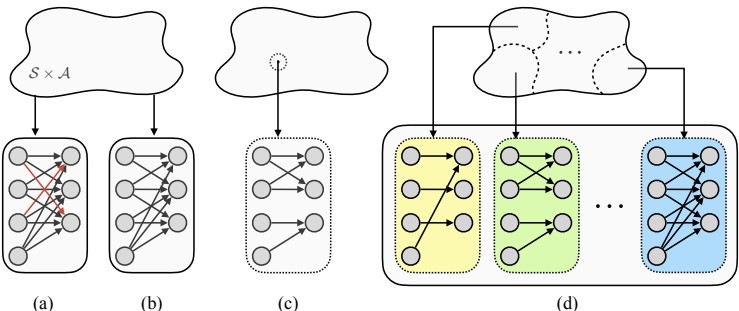

Figure 1: Comparison of different types of dynamics models. (a) Dense models employ the whole state and action variables for prediction. (b) Causal models examine conditional independences to discard unnecessary dependencies (red arrows in (a)). (c) Sample-specific approaches estimate variable dependencies on a per-sample basis. (d) Our model decomposes the data domain and infers fine-grained causal relationships on each event to use only locally relevant variables for prediction.

meaningful context for robust and fine-grained dynamics modeling. For this, we formulate the problem of finding a decomposition that maximizes the regularized maximum likelihood score and show that the optimal decomposition identifies a meaningful context that exhibits fine-grained causal relationships. A main challenge is that this involves three nested subtasks: discovering decomposition, examining local independences, and learning dynamics model. To this end, we propose a practical gradient-based method to learn a discrete latent codebook utilizing vector quantization, which enables the joint optimization differentiable, allowing efficient end-to-end training (Fig. 2). As a result, our method incorporates fine-grained causal relationships into dynamics modeling, leading to improved robustness in MBRL over prior causal dynamics models.

We empirically validate the effectiveness of our method on both discrete and continuous control environments. For the evaluation, we measure the performance of dynamics models on the downstream tasks that require fine-grained causal reasoning. Experimental results demonstrate the effectiveness of our method for fine-grained causal reasoning which improves robustness and generalization in MBRL. Detailed analysis of our method shows that it successfully discovers meaningful contexts and recovers fine-grained causal relationships.

## 2 PRELIMINARIES

We first briefly introduce the notations and terminologies used throughout the paper. Then, we examine related works on causal dynamics learning for RL and fine-grained causal relationships.

### 2.1 BACKGROUND

**Structural causal model.** We adopt a framework of a structural causal model (SCM) (Pearl, 2009) to understand the relationship among variables in transition dynamics. An SCM $\mathcal{M}$ is defined as a tuple $\langle \mathbf{V}, \mathbf{U}, \mathbf{F}, P(\mathbf{U}) \rangle$, where $\mathbf{V} = \{X_1, \cdots, X_d\}$ is a set of endogenous variables and $\mathbf{U}$ is a set of exogenous variables. A set of functions $\mathbf{F} = \{f_1, \cdots, f_d\}$ determine how each variable is generated; $X_j = f_j(Pa(j), \mathbf{U}_j)$ where $Pa(j) \subseteq \mathbf{V} \setminus \{X_j\}$ is parents of $X_j$ and $\mathbf{U}_j \subseteq \mathbf{U}$. An SCM $\mathcal{M}$ induces a directed acyclic graph (DAG) $\mathcal{G} = (V, E)$, i.e., a causal graph (CG) (Peters et al., 2017), where $V = \{1, \ldots, d\}$ and $E \subseteq V \times V$ are the set of nodes and edges, respectively. Each edge $(i, j) \in E$ denotes a direct causal relationship from $X_i$ to $X_j$. An SCM and its corresponding causal graph entail the conditional independence relationship of each variable (namely, local Markov property): $X_i \perp\!\!\!\perp ND(X_i) \mid Pa(X_i)$, where $ND(X_i)$ is a non-descendant of $X_i$.

**Factored Markov Decision Process.** A Markov Decision Process (MDP) (Sutton & Barto, 2018) is defined as a tuple $\langle \mathcal{S}, \mathcal{A}, T, r, \gamma \rangle$ where $\mathcal{S}$ is a state space, $\mathcal{A}$ is an action space, $T : \mathcal{S} \times \mathcal{A} \to \mathcal{P}(\mathcal{S})$ is a transition dynamics, $r$ is a reward function, and $\gamma$ is a discount factor. We consider a factored MDP (Kearns & Koller, 1999) where the state and action spaces are factorized as $\mathcal{S} = \mathcal{S}_1 \times \cdots \times \mathcal{S}_N$ and

$\mathcal{A} = \mathcal{A}_1 \times \cdots \times \mathcal{A}_M$, and a single-step transition dynamics is factorized as $p(s' \mid s, a) = \prod_j p(s'_j \mid s, a)$ where $s = (s_1, \cdots, s_N)$ and $a = (a_1, \cdots, a_M)$.

**Assumptions and notations.** We are concerned with an SCM associated with the transition dynamics in a factored MDP where we assume that states are fully observable. To properly identify the causal relationships in MBRL, we make assumptions standard in the field (Ding et al., 2022; Wang et al., 2021; 2022; Seitzer et al., 2021; Pitis et al., 2020; 2022), namely, Markov property (Pearl, 2009), faithfulness (Peters et al., 2017), causal sufficiency (Spirtes et al., 2000), and that causal connections only appear within consecutive time steps (i.e., $t \rightarrow t+1$). Throughout the paper, a causal graph $\mathcal{G} = (V, E)$ consists of the set of nodes $V = \mathbf{X} \cup \mathbf{Y}$ and the set of edges $E \subseteq \mathbf{X} \times \mathbf{Y}$, where $\mathbf{X} = \{S_1, \cdots, S_N, A_1, \cdots, A_M\}$ and $\mathbf{Y} = \{S'_1, \cdots, S'_N\}$. $Pa(j)$ denotes parent variables of $S'_j$. With these assumptions, the conditional independences

$$S'_j \perp\!\!\!\perp \mathbf{X} \setminus Pa(j) \mid Pa(j) \tag{1}$$

entailed by the causal graph faithfully represent the causal relationships between the variables and the transition dynamics is factorized as $p(s' \mid s, a) = \prod_j p(s'_j \mid s, a) = \prod_j p(s'_j \mid Pa(j))$.

**Dynamics modeling.** Traditional dynamics models use the whole state and action variables to predict the future state, i.e., modeling $\prod_j p(s'_j \mid s, a)$. Prior *causal* dynamics models (Wang et al., 2021; 2022; Ding et al., 2022) examine conditional independences to recover causal relationships and employ only parent variables for prediction, i.e., modeling $\prod_j p(s'_j \mid Pa(j))$. Consequently, causal dynamics models are more robust to unseen states by discarding unnecessary dependencies. In this work, we infer fine-grained causal relationships by discovering local independences and use potentially fewer dependencies for dynamics modeling, as shown in Fig. 1.

## 2.2 RELATED WORK

**Causal dynamics models in RL.** There is a growing body of literature on the intersection of causality and RL (De Haan et al., 2019; Buesing et al., 2019; Zhang et al., 2020a; Sontakke et al., 2021; Schölkopf et al., 2021; Zholus et al., 2022; Zhang et al., 2020b). One focus is causal dynamics learning, which aims to infer the causal structure of the underlying transition dynamics (Li et al., 2020; Yao et al., 2022; Bongers et al., 2018; Wang et al., 2022; Ding et al., 2022; Feng et al., 2022; Huang et al., 2022) (more broad literature of causal reasoning in RL is discussed in Appendix A.1). Given the explicit state and action variables in factored MDP, recent works utilize gradient-based causal discovery algorithm (Wang et al., 2021; Brouillard et al., 2020), conditional independence tests (Ding et al., 2022), or conditional mutual information (Wang et al., 2022) to infer the causal graph and train the dynamics model with the inferred causal graph by using only relevant variables for prediction. In contrast, our method infers fine-grained causal relationships by discovering local independences. Thus, our approach provides a more detailed understanding of the dynamics, leading to improved robustness and generalization over the prior causal dynamics models.

**Discovering fine-grained causal relationships.** The fine-grained causal relationships have been utilized to improve RL performance in various ways, e.g., with data augmentation (Pitis et al., 2022), efficient planning (Hoey et al., 1999; Chitnis et al., 2021), or exploration (Seitzer et al., 2021). Previous works exploited prior knowledge of them (Pitis et al., 2022), or leveraged the true dynamics model explicitly (Chitnis et al., 2021). Without those prior information, Pitis et al. (2020) devised an transformer-based model to estimate the variable dependencies for each sample by using attention score. Another line of work learn sparse and modular dynamics (Goyal et al., 2021c;b;a), which can be viewed as an implicit approach to discovering local independence relationships. In the field of causality, local independence relationship has been widely studied especially for discrete variables, e.g., context-specific independence (Boutilier et al., 2013; Zhang & Poole, 1999; Poole, 1998; Dal et al., 2018; Tikka et al., 2019) (see Appendix A.2 for the background on local independence). Recently, NCD (Hwang et al., 2023) proposed a gradient-based method to discover local independences allowing continuous variables. While it also infers local independences on a per-sample basis, our method infers local independences per event, i.e., subgroup of the data domain, which helps prevent overfitting to individual samples and allows more robust causal modeling.

## 3 FINE-GRAINED CAUSAL DYNAMICS LEARNING

We first describe a brief background on the local independences and local causal graph that represents the fine-grained causal relationships (Sec. 3.1). We then formulate a problem of finding the optimal decomposition and describe its implications (Sec. 3.2). As a practical approach, we present our proposed causal dynamics model that discovers decomposition and event-specific causal relationships with vector quantization, which enables joint differentiable optimization (Sec. 3.3). Finally, we provide a theoretical analysis of our approach to identifying the meaningful context that exhibits fine-grained causal relationships (Sec. 3.4).

### 3.1 LOCAL INDEPENDENCE AND LOCAL CAUSAL GRAPH

We first describe how local independence provides a way to understand fine-grained causal relationships between the variables. Analogous to the conditional independence in Eq. (1) explaining the causal relationship between the variables, local independence, which is written as:

$$S'_j \perp\!\!\!\perp \mathbf{X} \setminus Pa(j; \mathcal{E}) \mid Pa(j; \mathcal{E}), \mathcal{E}, \tag{2}$$

where $\mathcal{E} \subseteq \mathcal{X}$ is a subset of the joint state and action space $\mathcal{X} = \mathcal{S} \times \mathcal{A}$ and $Pa(j; \mathcal{E}) \subseteq Pa(j)$ is a minimal subset of $Pa(j)$ in which the local independence on $\mathcal{E}$ holds,[1] implies that only $Pa(j; \mathcal{E})$ are locally relevant variables for prediction on event $\mathcal{E}$ and the rest of the parent variables become redundant.

**Definition 1** (Local Causal Graph). *Local causal graph (LCG) on* $\mathcal{E} \subseteq \mathcal{X}$ *is* $\mathcal{G}_\mathcal{E} = (V, E_\mathcal{E})$ *where* $E_\mathcal{E} = \{(i, j) \mid i \in Pa(j; \mathcal{E})\}$.

Local causal graph $\mathcal{G}_\mathcal{E} \subseteq \mathcal{G}$ is a subgraph of the causal graph $\mathcal{G}$ which represents fine-grained causal relationships under the event $\mathcal{E}$. Clearly, $\mathcal{G}_\mathcal{X} = \mathcal{G}$. Also, $\mathcal{G}_\mathcal{E} \subsetneq \mathcal{G}$ does not always hold on any $\mathcal{E}$, and our goal is to find important contexts that entail a fine-grained causal relationship.

**Proposition 1** (Monotonicity). *Let* $\mathcal{F} \subseteq \mathcal{E}$. *Then,* $\mathcal{G}_\mathcal{F} \subseteq \mathcal{G}_\mathcal{E}$.

As the event we focus on becomes more specific (i.e., $\mathcal{F} \subseteq \mathcal{E}$), finer-grained relationships may arise (i.e., $\mathcal{G}_\mathcal{F} \subseteq \mathcal{G}_\mathcal{E}$), but it also becomes less likely to happen. Therefore, it is important to capture the context which is more likely (i.e., large $p(\mathcal{E})$), and more meaningful (i.e., sparse $\mathcal{G}_\mathcal{E}$).

### 3.2 SCORE FOR THE DECOMPOSITION AND GRAPHS

We consider a decomposition $\{\mathcal{E}_z\}_{z=1}^K$ of the domain $\mathcal{X}$ where $K$ is a small number which is a hyperparameter of our model. By decomposing the domain into a few subgroups, we aim to capture meaningful contexts that render sparse dependencies for robust and fine-grained dynamics modeling. It is worth noting that such events are not given as prior information.

For now, let us consider an arbitrary decomposition $\{\mathcal{E}_z\}_{z=1}^K$. We define a variable $Z$ representing the decomposition, defined as $Z = z$ if $(s, a) \in \mathcal{E}_z$ for all $z \in [K]$ (Hwang et al., 2023). For brevity, we denote $Pa(j; \mathcal{E}_z)$ as $Pa(j, z)$ and $\mathcal{G}_{\mathcal{E}_z}$ as $\mathcal{G}_z$. Each local independence $S'_j \perp\!\!\!\perp \mathbf{X} \setminus Pa(j, z) \mid Pa(j, z), \mathcal{E}_z$ is then equivalently written as $S'_j \perp\!\!\!\perp \mathbf{X} \setminus Pa(j, z) \mid Pa(j, z), Z = z$. The transition dynamics for each $S'_j$ can be written as:

$$p(s'_j \mid s, a) = \sum_z p(s'_j \mid s, a, z)p(z \mid s, a) = \sum_z p(s'_j \mid Pa(j, z), z)p(z \mid s, a), \tag{3}$$

where $p(z \mid s, a) = 1$ if $(s, a) \in \mathcal{E}_z$ otherwise 0. This illustrates our approach to fine-grained dynamics modeling, i.e., employing only locally relevant variables $Pa(j, z)$ for each $\mathcal{E}_z$. We now consider the following regularized maximum likelihood score:

$$\mathcal{S}(\{\mathcal{G}_z, \mathcal{E}_z\}_{z=1}^K) := \sup \mathbb{E}\left[\log \hat{p}(s' \mid s, a; \{\mathcal{G}_z, \mathcal{E}_z\}) - \lambda|\mathcal{G}_z|\right], \tag{4}$$

where $\{\mathcal{E}_z\}_{z=1}^K$ is the decomposition, $\mathcal{G}_z$ is the graph on each $\mathcal{E}_z$, and the dynamics model $\hat{p}$ uses $\mathcal{G}_z$ for each $\mathcal{E}_z$. It is worth noting that due to the nature of factored MDP where the causal graph is

---

[1] We provide a formal definition and detailed background of local independence in Appendix B.1.

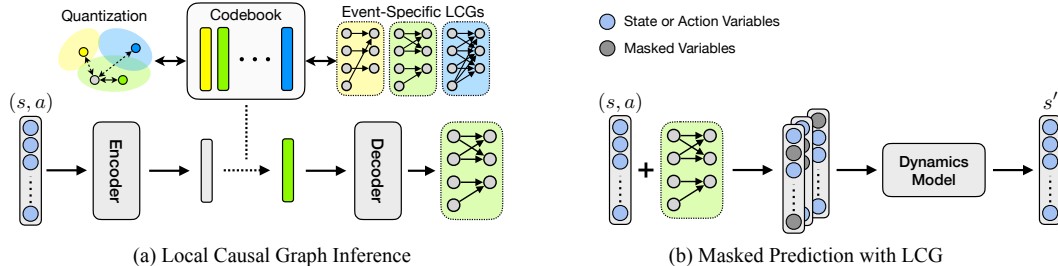

(a) Local Causal Graph Inference        (b) Masked Prediction with LCG

Figure 2: Overall framework. (a) For each sample $(s, a)$, our method infers the event to which the sample belongs through quantization, and the corresponding event-specific local causal graph (LCG) that represents fine-grained causal relationships. (b) Dynamics model is trained to predict the next state using only relevant variables based on the inferred LCG.

directed bipartite, each Markov equivalence class contains a single unique causal graph. Given this background, the causal graph is *uniquely identifiable* with oracle conditional independence test (Ding et al., 2022) or score maximization (Huang et al., 2018; Brouillard et al., 2020).

**Proposition 2.** *Let $\{\mathcal{E}_z\}_{z=1}^K$ be the arbitrary decomposition. Let $\{\hat{\mathcal{G}}_z\}_{z=1}^K$ be the graphs on each $\mathcal{E}_z$ that maximizes the score: $\{\hat{\mathcal{G}}_z\}_{z=1}^K \in \operatorname{argmax}_{\{\mathcal{G}_z\}} \mathcal{S}(\{\mathcal{G}_z, \mathcal{E}_z\}_{z=1}^K)$. With the Assumptions 1 to 4, each $\hat{\mathcal{G}}_z$ is true LCG on corresponding $\mathcal{E}_z$ for small enough $\lambda > 0$. In particular, if $K = 1$, then $\hat{\mathcal{G}} = \mathcal{G}$ where $\mathcal{G}$ is the ground truth causal graph.*

If $K = 1$, this degenerates to the prior score-based approach and would yield $\mathcal{G}$. On the other hand, any arbitrary decomposition of $K > 1$ also does not always provide a fine-grained understanding, e.g., $\mathcal{G}_z = \mathcal{G}$ for all $z$ in the worst case. Thus, we aim to discover the decomposition (and corresponding LCGs) that maximizes the score.

**Proposition 3.** *Let $\{\hat{\mathcal{G}}_z, \hat{\mathcal{E}}_z\}_{z=1}^K \in \operatorname{argmax}_{\{\mathcal{G}_z, \mathcal{E}_z\}} \mathcal{S}(\{\mathcal{G}_z, \mathcal{E}_z\}_{z=1}^K)$. Then, each $\hat{\mathcal{G}}_z$ is the true LCG on $\hat{\mathcal{E}}_z$ for all $z \in [K]$. Also, let $\{\mathcal{F}_z\}_{z=1}^K$ be the arbitrary decomposition and $\{\mathcal{G}_z\}_{z=1}^K$ be the corresponding true LCGs on each $\mathcal{F}_z$. Then, with the Assumptions 1 to 5, $\mathbb{E}\left[|\hat{\mathcal{G}}_z|\right] \leq \mathbb{E}\left[|\mathcal{G}_z|\right]$ holds for small enough $\lambda > 0$.*

This states that the decomposition that maximizes the score is optimal in terms of $\mathbb{E}\left[|\mathcal{G}_z|\right] = \sum_z p(\mathcal{E}_z)|\mathcal{G}_z|$, which implies that this captures the meaningful events that exhibit fine-grained causal relationships (i.e., sparse $\mathcal{G}_z$ with large $p(\mathcal{E}_z)$). We now proceed to describe our practical approach to find such decomposition for robust and fine-grained dynamics modeling in MBRL.

### 3.3 CAUSAL DYNAMICS LEARNING WITH QUANTIZED LOCAL INDEPENDENCE DISCOVERY

As described above, the decomposition and corresponding graphs that maximize the score (Eq. (4)), i.e., $\{\hat{\mathcal{G}}_z, \hat{\mathcal{E}}_z\}_{z=1}^K \in \operatorname{argmax}_{\{\mathcal{G}_z, \mathcal{E}_z\}} \mathcal{S}(\{\mathcal{G}_z, \mathcal{E}_z\}_{z=1}^K)$, provide a fine-grained understanding of the dynamics. Our goal is to (i) find decomposition (i.e., $\{\mathcal{E}_z\}$) which captures meaningful contexts, (ii) discover locally relevant variables $Pa(j, z)$ on each event (i.e., $\{\mathcal{G}_z\}$), and (iii) train the dynamics model $\hat{p}(s' \mid s, a)$ using them. However, this involves three nested subtasks, and naive optimization with respect to decomposition is generally intractable. To this end, we propose a practical gradient-based method which allows efficient joint optimization of three objectives. Our key idea is to learn a discrete latent codebook $C = \{e_z\}$ where each code $e_z$ represents $(\mathcal{G}_z, \mathcal{E}_z)$, i.e., the pair of event and corresponding graph, by utilizing vector quantization. The training of the codebook is differentiable and can be jointly trained with the dynamics model $\hat{p}$, resolving the challenging task of joint optimization of three objectives. The overall framework is illustrated in Fig. 2.

**Discrete latent codebook representing the decomposition.** First, with the encoder $g_{\mathrm{enc}}$, each sample $(s, a)$ is encoded into a latent embedding $h$, which is then quantized to the nearest prototype vector (i.e., code) in the codebook $C = \{e_1, \cdots, e_K\}$, following (Van Den Oord et al., 2017) as:

$$e = e_z, \quad \text{where} \quad z = \operatorname*{argmin}_{j \in [K]} \|h - e_j\|_2. \tag{5}$$

The quantization entails the decomposition of $\mathcal{X}$ since each sample corresponds to exactly one of the latent codes. In other words, the discrete latent codebook represents the decomposition as $\mathcal{E}_z = \{(s, a) \mid e = e_z\}$ for all $z \in [K]$. The quantization corresponds to the term $p(z \mid s, a)$ in Eq. (3) as: $p(z \mid s, a) = 1$ if $e = e_z$ and otherwise 0, i.e., determines the event to which the sample belongs. Thus, the codebook $C = \{e_z\}_{z=1}^K$ serves as a proxy for decomposition $\{\mathcal{E}_z\}_{z=1}^K$.

**Discrete latent codebook representing the local independences.** Each code $e_z$ is then decoded to an adjacency matrix $A_z \in \{0, 1\}^{(N+M) \times N}$ that represents the inferred graph $\mathcal{G}_z$. In particular, the output of the decoder $g_{\text{dec}}$ is the parameters of Bernoulli distributions from which adjacency matrix is sampled: $A \sim g_{\text{dec}}(e) = [p_{ij}]$. To properly backpropagate gradients, we adopt Gumbel-Softmax reparametrization trick (Jang et al., 2016; Maddison et al., 2016).

**Dynamics model learning.** The dynamics model employs only relevant variables for prediction with respect to the adjacency matrix $A$ (Fig. 2 (b)). Specifically, $\log \hat{p}(s' \mid s, a; A) = \sum_j \log \hat{p}(s'_j \mid s, a; A_j)$, where $A_j$ is the $j$-th column of $A$. Each entry of $A_j \in \{0, 1\}^{(N+M)}$ indicates whether the corresponding state or action variable will be used or not to determine the next state $s'_j$. This corresponds to the term $p(s'_j \mid Pa(j, z), z)$ in Eq. (3). For the implementation of $\hat{p}(s'_j \mid s, a; A_j)$, we mask out the features of unused variables (Brouillard et al., 2020).

**Training objective.** We employ a regularization loss $\lambda \cdot \|A\|_1$ to induce a sparse LCG, where $\lambda$ is a hyperparameter. The masked prediction loss with the regularization is as follows:

$$\mathcal{L}_{\text{pred}} = -\log \hat{p}(s' \mid s, a; A) + \lambda \cdot \|A\|_1. \tag{6}$$

To update the codebook, we use a quantization loss (Van Den Oord et al., 2017):

$$\mathcal{L}_{\text{quant}} = \|\text{sg}\,[h] - e\|_2^2 + \beta \cdot \|h - \text{sg}\,[e]\|_2^2, \tag{7}$$

where $\text{sg}\,[\cdot]$ is a stop-gradient operator and $\beta$ is a hyperparameter. The first term is the codebook loss which moves each code toward the center of the embeddings assigned to it. The second term is the commitment loss which encourages the quantization encoder outputs the embeddings close to the prototype vectors. The resulting training objective is $\mathcal{L}_{\text{total}} = \mathcal{L}_{\text{pred}} + \mathcal{L}_{\text{quant}}$.

### 3.4 Identifiability and discussions

So far, we have described how our method learns the decomposition and LCGs through the discrete latent codebook $C$ as a proxy where each code $e_z$ corresponds to the pair of event and graph $(\mathcal{E}_z, \mathcal{G}_z)$, and joint training of the dynamics model $\hat{p}$ and the codebook $C$. Considering that $\mathcal{L}_{\text{pred}}$ corresponds to the score $\mathcal{S}(\{\mathcal{G}_z, \mathcal{E}_z\}_{z=1}^K)$ in Eq. (4) and $\mathcal{L}_{\text{quant}}$ is a mean squared error in the latent space which can be minimized to 0, our method is a practical approach to the score maximization which allows efficient end-to-end training. Finally, we show the identifiability that the optimal decomposition that maximizes the score identifies a meaningful context that exhibits fine-grained causal relationships.

**Theorem 1** (Identifiability). *Let $\{\hat{\mathcal{G}}_z, \hat{\mathcal{E}}_z\}_{z=1}^K \in \text{argmax}_{\{\mathcal{G}_z, \mathcal{E}_z\}} \mathcal{S}(\{\mathcal{G}_z, \mathcal{E}_z\}_{z=1}^K)$ and $K > 1$. Let $\mathcal{D} \subsetneq \mathcal{X}$ where $\mathcal{G}_\mathcal{D} \subsetneq \mathcal{G}$. Suppose $\mathcal{G}_\mathcal{F} = \mathcal{G}_\mathcal{D}$ for any $\mathcal{F} \subseteq \mathcal{D}$, and $\mathcal{G}_\mathcal{F} = \mathcal{G}$ for any $\mathcal{F} \subseteq \mathcal{D}^c$. Then, with the Assumptions 1 to 5, there exists $I \subsetneq [K]$ such that (i) $\bigcup_{i \in I} \hat{\mathcal{E}}_i \subseteq \mathcal{D}$ and $p(\mathcal{D} \setminus \bigcup_{i \in I} \hat{\mathcal{E}}_i) = 0$, (ii) $\hat{\mathcal{G}}_z = \mathcal{G}_\mathcal{D}$ for all $z \in I$ and $\hat{\mathcal{G}}_z = \mathcal{G}$ for all $z \notin I$.*

It states that the optimal decomposition $\{\hat{\mathcal{E}}_z\}_{z \in [K]}$ would discover the meaningful context $\mathcal{D}$ which exhibits fine-grained causal relationships, in the sense that events $\{\hat{\mathcal{E}}_z\}_{z \in I}$ identify $\mathcal{D}$ almost surely where $I \subsetneq [K]$. Thm. 1 implies that any choice of $K > 1$ would lead to the identification of meaningful context. In our method, the codebook size $K$ represents to the size of the decomposition, and in practice, we found that our method works reasonably well for any choice of $K > 1$. Omitted proofs are provided in Appendix B.2.

## 4 Experiments

In this section, we evaluate our method to examine the following questions: (1) Does our dynamics model improves robustness and generalization of MBRL? (Tables 1 and 2); (2) Does our method capture a meaningful context and fine-grained causal relationships? (Figs. 4 and 5); and (3) How does the choice of $K$ affect performance? (Table 3)

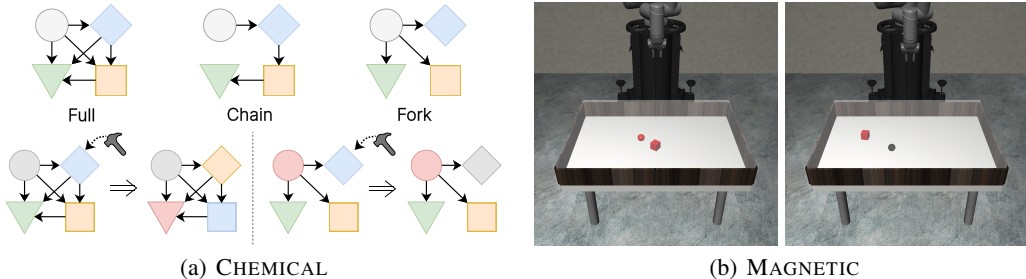

(a) CHEMICAL                           (b) MAGNETIC

Figure 3: Illustrations for each environment. (a) In Chemical, colors change by the action according to the underlying causal graph. (b) In Magnetic, the red object exhibits magnetism. (b-left) The box attracts the ball via magnetic force. (b-right) The box does not have an influence on the ball.

## 4.1 EXPERIMENTAL SETUP

The environments are designed to exhibit fine-grained causal relationships on a particular context, and explicit state variables are given as the observation to the agent (e.g., position, velocity), following prior works (Ding et al., 2022; Wang et al., 2022; Seitzer et al., 2021; Pitis et al., 2020; 2022). Detailed descriptions for each environment and setup are provided in Appendix C.1.

### 4.1.1 ENVIRONMENTS

**Chemical.** In the Chemical (Ke et al., 2021) environment, there are 10 nodes where each node is colored with one of 5 colors, and an action is setting the color of a node. According to the underlying causal graph, an action changes the colors of the intervened object's descendants as depicted in Fig. 3(a). The task is to match the colors of each node to the given target. We design two settings, named *full-fork* and *full-chain*: the underlying causal graph is both *full*, and the local causal graph is *fork* and *chain*, respectively. For example, in *full-fork*, local causal graph *fork* corresponds to the context where the color of the root node is red. In other words, for each node, all other parent nodes except the root become irrelevant according to the particular color of the root node. During the test, the root color is set to activate local causal graph. Here, the agent receives a noisy observation for some nodes (except the root) and the task is to match the colors of other observable nodes. The agent capable of fine-grained causal reasoning would generalize well since corrupted nodes are locally spurious to infer the colors of other nodes, as depicted in Appendix C.1 (Fig. 6).

**Magnetic.** We design the robot arm manipulation environment based on the Robosuite framework (Zhu et al., 2020). There is a moving ball and a box on the table, colored red or black (Fig. 3(b)). Red color indicates that the object is *magnetic*, and attracts the other magnetic object. For example, when they are both colored red, magnetic force will be applied, and the ball will move toward the box. Otherwise, the box would have no influence on the ball. The task is to move the robot arm to reach the ball predicting its trajectory. The color and position of the objects are randomly initialized for each episode. During the test, one of the objects is black, and the box is located at an unseen position. Since the position of the box is unnecessary for predicting the movement of the ball under non-magnetic context, the agent aware of the fine-grained causal relationships would generalize well to unseen out-of-distribution (OOD) states.

### 4.1.2 BASELINES

We first consider traditional **dense models**, i.e., a monolithic network implemented as multi-layer perceptron (MLP) which approximates the dynamics $p(s' \mid s, a)$, and a modular network which has a separate network for each variable: $\prod_j p(s'_j \mid s, a)$. In addition, we include a graph neural network (GNN) (Kipf et al., 2020) and NPS (Goyal et al., 2021a). GNN learns the relational information between variables and NPS learns sparse and modular dynamics. **Causal models**, including CDL (Wang et al., 2022) and GRADER (Ding et al., 2022), infer causal relationships between the variables for dynamics learning: $\prod_j p(s'_j \mid Pa(j))$, utilizing conditional mutual information and conditional independence test, respectively. We also include an *oracle* causal model, which leverages the ground truth (global) causal graph. Finally, we compare to a **local causal model**, NCD (Hwang et al., 2023), which infers the variable dependencies for each sample.

Table 1: Average episode reward on training and downstream tasks in each environment. In Chemical, $n$ denotes the number of noisy nodes in downstream tasks.

| Methods | Chemical (*full-fork*) | | | | Chemical (*full-chain*) | | | | Magnetic | |
|---|---|---|---|---|---|---|---|---|---|---|
| | Train ($n=0$) | Test ($n=2$) | Test ($n=4$) | Test ($n=6$) | Train ($n=0$) | Test ($n=2$) | Test ($n=4$) | Test ($n=6$) | Train | Test |
| MLP | $19.00_{\pm0.83}$ | $6.49_{\pm0.48}$ | $5.93_{\pm0.71}$ | $6.84_{\pm1.17}$ | $17.91_{\pm0.87}$ | $7.39_{\pm0.65}$ | $6.63_{\pm0.58}$ | $6.78_{\pm0.93}$ | $8.37_{\pm0.74}$ | $0.86_{\pm0.45}$ |
| Modular | $18.55_{\pm1.00}$ | $6.05_{\pm0.70}$ | $5.65_{\pm0.50}$ | $6.43_{\pm1.00}$ | $17.37_{\pm1.63}$ | $6.61_{\pm0.63}$ | $7.01_{\pm0.55}$ | $7.04_{\pm1.07}$ | $8.45_{\pm0.80}$ | $0.88_{\pm0.52}$ |
| GNN | $18.60_{\pm1.19}$ | $6.61_{\pm0.92}$ | $6.15_{\pm0.74}$ | $6.95_{\pm0.78}$ | $16.97_{\pm1.85}$ | $6.89_{\pm0.28}$ | $6.38_{\pm0.28}$ | $6.56_{\pm0.53}$ | $8.53_{\pm0.83}$ | $0.92_{\pm0.51}$ |
| NPS | $7.71_{\pm1.22}$ | $5.82_{\pm0.83}$ | $5.75_{\pm0.57}$ | $5.54_{\pm0.80}$ | $8.20_{\pm0.54}$ | $6.92_{\pm1.03}$ | $6.88_{\pm0.79}$ | $6.80_{\pm0.39}$ | $3.13_{\pm1.00}$ | $0.91_{\pm0.69}$ |
| CDL | $18.95_{\pm1.40}$ | $9.37_{\pm1.33}$ | $8.23_{\pm0.40}$ | $9.50_{\pm1.18}$ | $17.95_{\pm0.83}$ | $8.71_{\pm0.55}$ | $8.65_{\pm0.38}$ | $10.23_{\pm0.50}$ | $\mathbf{8.75}_{\pm0.69}$ | $1.10_{\pm0.67}$ |
| GRADER | $18.65_{\pm0.98}$ | $9.27_{\pm1.31}$ | $8.79_{\pm0.65}$ | $10.61_{\pm1.31}$ | $17.71_{\pm0.54}$ | $8.69_{\pm0.56}$ | $8.75_{\pm0.80}$ | $10.14_{\pm0.33}$ | - | - |
| Oracle | $\mathbf{19.64}_{\pm1.18}$ | $7.83_{\pm0.87}$ | $8.04_{\pm0.62}$ | $9.66_{\pm0.21}$ | $17.79_{\pm0.76}$ | $8.47_{\pm0.69}$ | $8.85_{\pm0.78}$ | $10.29_{\pm0.37}$ | $8.42_{\pm0.86}$ | $0.95_{\pm0.55}$ |
| NCD | $19.30_{\pm0.95}$ | $10.95_{\pm1.63}$ | $9.11_{\pm0.63}$ | $10.32_{\pm0.93}$ | $\mathbf{18.27}_{\pm0.27}$ | $9.60_{\pm1.52}$ | $8.86_{\pm0.23}$ | $10.32_{\pm0.37}$ | $8.48_{\pm0.70}$ | $1.31_{\pm0.77}$ |
| Ours | $19.28_{\pm0.87}$ | $\mathbf{15.27}_{\pm2.53}$ | $\mathbf{14.73}_{\pm1.68}$ | $\mathbf{13.62}_{\pm2.56}$ | $17.22_{\pm0.61}$ | $\mathbf{13.36}_{\pm3.60}$ | $\mathbf{12.35}_{\pm3.23}$ | $\mathbf{12.00}_{\pm1.21}$ | $8.52_{\pm0.74}$ | $\mathbf{4.81}_{\pm3.01}$ |

Table 2: Prediction accuracy on ID ($n=0$) and OOD ($n=2, 4, 6$) states in Chemical environment.

| Setting | | MLP | Modular | GNN | NPS | CDL | GRADER | Oracle | NCD | Ours |
|---|---|---|---|---|---|---|---|---|---|---|
| *full-fork* | ($n=0$) | $88.31_{\pm1.58}$ | $89.24_{\pm1.52}$ | $88.81_{\pm1.44}$ | $58.34_{\pm2.08}$ | $89.22_{\pm1.67}$ | $87.75_{\pm1.64}$ | $89.63_{\pm1.62}$ | $\mathbf{90.07}_{\pm1.22}$ | $89.46_{\pm1.40}$ |
| | ($n=2$) | $31.11_{\pm1.69}$ | $26.53_{\pm3.45}$ | $36.29_{\pm3.45}$ | $40.56_{\pm4.61}$ | $35.59_{\pm1.85}$ | $37.93_{\pm1.06}$ | $33.87_{\pm1.34}$ | $41.60_{\pm5.08}$ | $\mathbf{66.44}_{\pm12.22}$ |
| | ($n=4$) | $30.44_{\pm2.28}$ | $24.73_{\pm5.61}$ | $25.80_{\pm3.48}$ | $26.81_{\pm4.37}$ | $35.82_{\pm1.40}$ | $38.94_{\pm1.63}$ | $36.48_{\pm1.80}$ | $37.47_{\pm2.13}$ | $\mathbf{58.49}_{\pm10.20}$ |
| | ($n=6$) | $32.39_{\pm1.76}$ | $26.73_{\pm8.31}$ | $21.58_{\pm3.44}$ | $23.02_{\pm4.27}$ | $42.22_{\pm1.39}$ | $45.74_{\pm2.25}$ | $42.47_{\pm0.75}$ | $42.27_{\pm1.82}$ | $\mathbf{49.09}_{\pm4.77}$ |
| *full-chain* | ($n=0$) | $84.38_{\pm1.31}$ | $85.92_{\pm1.15}$ | $85.41_{\pm1.84}$ | $58.48_{\pm2.81}$ | $\mathbf{86.85}_{\pm1.47}$ | $84.24_{\pm1.22}$ | $85.76_{\pm1.56}$ | $85.63_{\pm1.01}$ | $86.07_{\pm1.62}$ |
| | ($n=2$) | $28.66_{\pm3.65}$ | $25.24_{\pm4.68}$ | $29.22_{\pm3.39}$ | $38.73_{\pm2.63}$ | $34.90_{\pm1.59}$ | $36.82_{\pm3.12}$ | $34.63_{\pm1.78}$ | $40.04_{\pm6.21}$ | $\mathbf{60.34}_{\pm12.10}$ |
| | ($n=4$) | $26.52_{\pm4.26}$ | $24.94_{\pm4.81}$ | $23.28_{\pm4.98}$ | $27.69_{\pm4.28}$ | $36.52_{\pm1.72}$ | $37.41_{\pm2.84}$ | $38.31_{\pm2.48}$ | $37.47_{\pm2.98}$ | $\mathbf{56.64}_{\pm9.40}$ |
| | ($n=6$) | $24.15_{\pm4.17}$ | $25.09_{\pm5.91}$ | $20.53_{\pm6.96}$ | $24.45_{\pm3.84}$ | $42.06_{\pm1.29}$ | $43.48_{\pm4.14}$ | $42.87_{\pm2.08}$ | $41.19_{\pm1.66}$ | $\mathbf{53.29}_{\pm6.63}$ |

**Planning algorithm.** For all baselines and our method, we use a model predictive control (MPC) (Camacho & Alba, 2013) which selects the actions based on the prediction of the learned dynamics model. Specifically, we use the cross-entropy method (CEM) (Rubinstein & Kroese, 2004), which iteratively generates and optimizes action sequences.

**Implementation.** For the implementation of our method, we set the codebook size of $K = 16$, the regularization coefficient $\lambda = 0.001$, and the commitment coefficient $\beta = 0.25$ in all experiments. For the oracle causal model, we used the same network architecture as ours, to isolate the effect of learning fine-grained causal relationships. All methods have a similar number of model parameters for a fair comparison. For the evaluation, we ran 10 test episodes for every 40 training episodes. The results are averaged over eight different runs. Implementation details are provided in Appendix C.2. Learning curves for all downstream tasks with additional experimental results are shown in Appendix C.3.

## 4.2 DOWNSTREAM TASK PERFORMANCE

Table 1 demonstrate the downstream task performance of our method and baselines. While all the methods show similar performance in training, dense models suffer from OOD states in the downstream tasks. Causal models are generally more robust compared to dense models, as they infer the causal graph to discard irrelevant dependencies. Our method significantly outperforms the baselines in all downstream tasks, which implies that our method is capable of fine-grained causal reasoning and generalize well to unseen states. To investigate the robustness of our method in downstream tasks, we examine the prediction accuracy on in-distribution (ID) states, and OOD states in Chemical environment (Table 2). While all methods show similar prediction accuracy on ID states, the baselines show a significant performance drop under the presence of noisy variables, merely above 20% which is an expected accuracy of random guessing. In contrast, our method consistently outperforms the baselines by a large margin, which implies that it successfully captures the fine-grained causal relationships. These results demonstrates the effectiveness of our method for fine-grained causal reasoning which improves robustness and generalization in MBRL.

## 4.3 DETAILED ANALYSIS

**Inferred LCGs.** We closely examine the discovered local causal graphs in Chemical (*full-fork*) to better understand our model's behavior (Fig. 4). Recall each code corresponds to the pair of event and LCG, we observe that all codes are used during training (Fig. 4(a)), but only a few of them are exploited during the test (Fig. 4(b)). The most commonly inferred LCG on OOD states during the

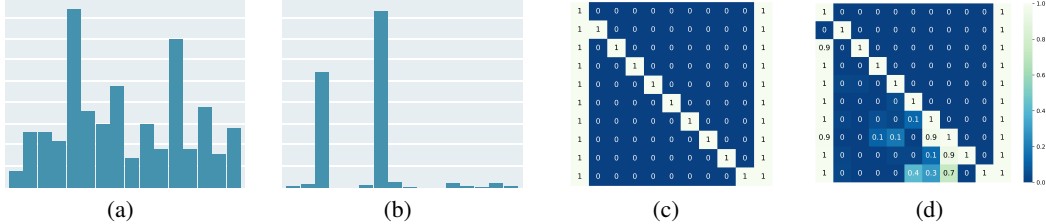

Figure 4: (a,b) Codebook histogram on (a) ID states during training and (b) OOD states during the test in Chemical (*full-fork*). (c) True causal graph of the *fork* structure. (d) Learned LCG corresponding to the most used code in (b).

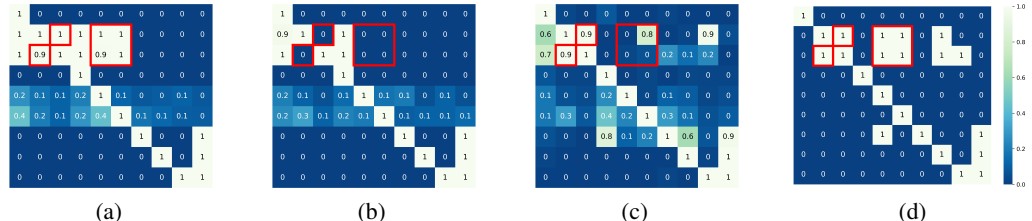

Figure 5: (a) Causal graph identified by our method in Magnetic. Red boxes indicate locally irrelevant edges under the non-magnetic event. (b-d) LCGs corresponding to the non-magnetic event inferred by (b) our method, and NCD on (c) ID and (d) OOD state.

test corresponds to a *fork* structure, as shown in Figs. 4(c) and 4(d). This implies that our method successfully identifies the meaningful context that exhibits fine-grained causal relationships.

**Comparison with sample-specific approach.** In Fig. 5, we examine the LCGs to further analyze the improved robustness of our approach compared to a sample-specific approach that does not explicitly capture meaningful events. As shown in Fig. 5(b), our method learns a proper LCG during training, and the inference is consistent in both ID and OOD states on non-magnetic context. In contrast, as shown in Figs. 5(c) and 5(d), the inference of sample-specific approach is inconsistent between ID and OOD states on the same non-magnetic context. This is because sample-specific approach infers the variable dependencies for each sample, and this incurs overfitting which makes the inference on OOD states inconsistent. As opposed to sample-specific approach, our method learns an LCG for each event, leading to the robust inference on OOD states.

Table 3: Ablation on codebook size.

| Methods | Chemical (*full-fork*) | | |
| --- | --- | --- | --- |
| | $(n = 2)$ | $(n = 4)$ | $(n = 6)$ |
| CDL | $9.37_{\pm1.33}$ | $8.23_{\pm0.40}$ | $9.50_{\pm1.18}$ |
| Oracle | $7.83_{\pm0.87}$ | $8.04_{\pm0.62}$ | $9.66_{\pm0.21}$ |
| NCD | $10.95_{\pm1.63}$ | $9.11_{\pm0.63}$ | $10.32_{\pm0.93}$ |
| Ours ($K = 2$) | $13.44_{\pm5.41}$ | $12.86_{\pm5.58}$ | $12.99_{\pm5.27}$ |
| Ours ($K = 4$) | $15.73_{\pm4.13}$ | $16.50_{\pm3.40}$ | $12.40_{\pm2.81}$ |
| Ours ($K = 8$) | $14.95_{\pm1.16}$ | $15.03_{\pm2.61}$ | $13.42_{\pm2.67}$ |
| Ours ($K = 16$) | $15.27_{\pm2.53}$ | $14.73_{\pm1.68}$ | $13.62_{\pm2.56}$ |
| Ours ($K = 32$) | $16.12_{\pm1.43}$ | $14.35_{\pm1.37}$ | $14.79_{\pm2.13}$ |

**Ablation on the codebook size.** We first recall that the codebook size $K$ represents the size of the decomposition. As shown in Table 3, our method works reasonably well with any choice of $K$ and consistently outperforms baselines. Finally, as we described earlier, our method with $K = 1$ degenerates to prior causal dynamics model and cannot capture fine-grained causal relationships. This is shown in Fig. 5(a) that our method with $K = 1$ recovers the causal graph including locally irrelevant edges under the non-magnetic context.

## 5 CONCLUSION

We presented a novel approach to causal dynamics learning that infers fine-grained causal relationships, improving robustness and generalization of MBRL compared to previous approaches. We show that the decomposition that maximizes the score identifies the meaningful context existing in the system. As a practical approach, our method learns a discrete latent variable that represents the pairs of event and event-specific causal relationships by utilizing vector quantization. Compared to prior approaches, our method provides a fine-grained understanding of the dynamics and allows robust causal dynamics modeling by capturing event-specific causal relationships. We discuss limitations and future works in Appendix C.3.5.

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

# A APPENDIX FOR PRELIMINARY

## A.1 EXTENDED RELATED WORK

Recently, incorporating causal reasoning into RL has gained much attention in the community in various aspects. For example, causality has been shown to improve off-policy evaluation (Buesing et al., 2019; Oberst & Sontag, 2019), goal-directed tasks (Nair et al., 2019), credit assignment (Mesnard et al., 2021), robustness (Lyle et al., 2021; Volodin et al., 2020), policy transfer (Killian et al., 2022), explainability (Madumal et al., 2020), and policy learning with counterfactual data augmentation (Lu et al., 2020; Pitis et al., 2020; 2022). Causality has also been integrated with bandits (Bareinboim et al., 2015; Lee & Bareinboim, 2018) or imitation learning (Bica et al., 2021; De Haan et al., 2019; Zhang et al., 2020b) to handle the unobserved confounders and learn generalizable policies. Another line of work focused on causal reasoning over the high-dimensional visual observation (Lu et al., 2018; Feng et al., 2022; Rezende et al., 2020) where the representation learning is crucial (Zhang et al., 2019; Sontakke et al., 2021; Tomar et al., 2021). Our work falls into the category of improving dynamics learning by incorporating causality, where recent works have focused on the discovery of the causal relationships between the variables explicitly (Wang et al., 2021; 2022; Ding et al., 2022). On the contrary, our work incorporates fine-grained local causal relationships into dynamics learning, which is underexplored in prior works.

## A.2 BACKGROUND ON LOCAL INDEPENDENCE RELATIONSHIP

In this subsection, we provide the background on local independence relationship. We first describe context-specific independence (CSI) (Boutilier et al., 2013), which denotes a variable being conditionally independent of others given a particular context, not the full set of parents in the graph.

**Definition 2** (Context-Specific Independence (CSI) (Boutilier et al., 2013), reproduced from Hwang et al. (2023)). *$Y$ is said to be **contextually independent** of $\mathbf{X}_B$ given the context $\mathbf{X}_A = \mathbf{x}_A$ if $P\left(y \mid \mathbf{x}_A, \mathbf{x}_B\right) = P\left(y \mid \mathbf{x}_A\right)$, holds for all $y \in \mathcal{Y}$ and $\mathbf{x}_B \in \mathcal{X}_B$ whenever $P\left(\mathbf{x}_A, \mathbf{x}_B\right) > 0$. This will be denoted by $Y \perp\!\!\!\perp \mathbf{X}_B \mid \mathbf{X}_A = \mathbf{x}_A$.*

CSI has been widely studied especially for discrete variables with low cardinality, e.g., binary variables. Context-set specific independence (CSSI) generalizes the notion of CSI allowing continuous variables.

**Definition 3** (Context-Set Specific Independence (CSSI) (Hwang et al., 2023)). *Let $\mathbf{X} = \{X_1, \cdots, X_d\}$ be a non-empty set of the parents of $Y$ in a causal graph, and $\mathcal{E} \subseteq \mathcal{X}$ be an event with a positive probability. $\mathcal{E}$ is said to be a **context set** which induces **context-set specific independence (CSSI)** of $\mathbf{X}_{A^c}$ from $Y$ if $p\left(y \mid \mathbf{x}_{A^c}, \mathbf{x}_A\right) = p\left(y \mid \mathbf{x}'_{A^c}, \mathbf{x}_A\right)$ holds for every $\left(\mathbf{x}_{A^c}, \mathbf{x}_A\right), \left(\mathbf{x}'_{A^c}, \mathbf{x}_A\right) \in \mathcal{E}$. This will be denoted by $Y \perp\!\!\!\perp \mathbf{X}_{A^c} \mid \mathbf{X}_A, \mathcal{E}$.*

Intuitively, it denotes that the conditional distribution $p(y \mid x) = p(y \mid x_{A^c}, x_A)$ is the same for different values of $\mathbf{x}_{A^c}$, for all $x = (\mathbf{x}_{A^c}, \mathbf{x}_A) \in \mathcal{E}$. In other words, only a subset of the parent variables is sufficient for modeling $p(y \mid x)$ when restricted in $\mathcal{E}$.

# B APPENDIX FOR THEORETICAL ANALYSIS

## B.1 PRELIMINARIES

Now, we formally define local independence by adapting CSSI to our setting. As mentioned in Sec. 2, we consider factored MDP where the causal graph is directed bipartite. Note that $\mathbf{X} = \{S_1, \cdots, S_N, A_1, \cdots, A_M\}$, $\mathbf{Y} = \{S'_1, \cdots, S'_N\}$, and $Pa(j)$ is parent variables of $S'_j$.

**Assumption 1.** *We assume Markov property (Pearl, 2009), faithfulness (Peters et al., 2017), and causal sufficiency (Spirtes et al., 2000).*

We note that these assumptions are standard in the field to properly identify the causal relationships in MBRL (Ding et al., 2022; Wang et al., 2021; 2022; Seitzer et al., 2021; Pitis et al., 2020; 2022).

**Definition 4** (Local Independence). *Let $\mathbf{T} \subseteq Pa(j)$ and $\mathcal{E} \subseteq \mathcal{X}$ with $p(\mathcal{E}) > 0$. We say the local independence $S'_j \perp\!\!\!\perp \mathbf{X} \setminus \mathbf{T} \mid \mathbf{T}, \mathcal{E}$ holds on $\mathcal{E}$ if $p(s'_j \mid \mathbf{x}_{T^c}, \mathbf{x}_T) = p(s'_j \mid \mathbf{x}'_{T^c}, \mathbf{x}_T)$ holds for every $\left(\mathbf{x}_{T^c}, \mathbf{x}_T\right), \left(\mathbf{x}'_{T^c}, \mathbf{x}_T\right) \in \mathcal{E}.$*[2]

Local independence extends conditional independence, i.e., if conditional independence $S'_j \perp\!\!\!\perp \mathbf{X} \setminus \mathbf{T} \mid \mathbf{T}$ holds, then local independence $S'_j \perp\!\!\!\perp \mathbf{X} \setminus \mathbf{T} \mid \mathbf{T}, \mathcal{E}$ holds for any $\mathcal{E} \subseteq \mathcal{X}$. Local independence implies that only subset of the parent variables is locally relevant on $\mathcal{E}$, and any other remaining parent variables are locally irrelevant. Throughout the paper, we are concerned with the events with the positive probability, i.e., $p(\mathcal{E}) > 0$.

**Definition 5** (Local Parent Set). *$Pa(j; \mathcal{E})$ is a subset of $Pa(j)$ such that $S'_j \perp\!\!\!\perp \mathbf{X} \setminus Pa(j; \mathcal{E}) \mid Pa(j; \mathcal{E}), \mathcal{E}$ holds and $S'_j \not\perp\!\!\!\perp \mathbf{X} \setminus \mathbf{T} \mid \mathbf{T}, \mathcal{E}$ for any $\mathbf{T} \subsetneq Pa(j; \mathcal{E})$.*

In other words, $Pa(j; \mathcal{E})$ is a minimal subset of $Pa(j)$ in which the local independence on $\mathcal{E}$ holds. Clearly, $Pa(j; \mathcal{X}) = Pa(j)$, i.e., local independence on $\mathcal{X}$ is equivalent to the (global) conditional independence.

**Definition 1** (Local Causal Graph). *Local causal graph (LCG) on $\mathcal{E} \subseteq \mathcal{X}$ is $\mathcal{G}_\mathcal{E} = (V, E_\mathcal{E})$ where $E_\mathcal{E} = \{(i, j) \mid i \in Pa(j; \mathcal{E})\}$.*

Local causal graph is a subgraph of the causal graph, i.e., $\mathcal{G}_\mathcal{E} \subseteq \mathcal{G}$, which describes fine-grained causal relationships that arise under the event $\mathcal{E}$. Note that $\mathcal{G}_\mathcal{X} = \mathcal{G}$, i.e., local independence and LCG under $\mathcal{X}$ are equivalent to conditional independence and CG, respectively. Analogous to the faithfulness assumption (Peters et al., 2017) that no conditional independences other than ones entailed by CG are present, we introduce a similar assumption for LCG and local independence.

**Assumption 2** ($\mathcal{E}$-Faithfulness). *For any $\mathcal{E}$, no local independences on $\mathcal{E}$ other than the ones entailed by $\mathcal{G}_\mathcal{E}$ are present, i.e., for any $j$, there does not exists any $\mathbf{T}$ such that $Pa(j; \mathcal{E}) \setminus \mathbf{T} \neq \emptyset$ and $S'_j \perp\!\!\!\perp \mathbf{X} \setminus \mathbf{T} \mid \mathbf{T}, \mathcal{E}$.*

Regardless of the $\mathcal{E}$-faithfulness assumption, LCG always exists because $Pa(j; \mathcal{E})$ always exists. However, such LCG may not be unique. $\mathcal{E}$-faithfulness implies the uniqueness of $Pa(j; \mathcal{E})$ and $\mathcal{G}_\mathcal{E}$. See (Hwang et al., 2023, Example. 2) for the violation of $\mathcal{E}$-faithfulness. We now provide a proof of Prop. 1.

**Lemma 1** (Hwang et al. (2023), Prop. 4). *$S'_j \perp\!\!\!\perp \mathbf{X} \setminus Pa(j; \mathcal{E}) \mid Pa(j; \mathcal{E}), \mathcal{F}$ holds for any $\mathcal{F} \subseteq \mathcal{E}$.*

**Proposition 1** (Monotonicity). *Let $\mathcal{F} \subseteq \mathcal{E}$. Then, $\mathcal{G}_\mathcal{F} \subseteq \mathcal{G}_\mathcal{E}$.*

*Proof.* Since $S'_j \perp\!\!\!\perp \mathbf{X} \setminus Pa(j; \mathcal{E}) \mid Pa(j; \mathcal{E}), \mathcal{F}$ holds by Lemma 1, $Pa(j; \mathcal{F}) \subseteq Pa(j; \mathcal{E})$ holds by definition; otherwise, $Pa(j; \mathcal{F}) \setminus Pa(j; \mathcal{E}) \neq \emptyset$ which leads to contradiction. Therefore, $Pa(j; \mathcal{F}) \subseteq Pa(j; \mathcal{E})$ for all $j$ and thus $\mathcal{G}_\mathcal{F} \subseteq \mathcal{G}_\mathcal{E}$. $\qquad\square$

### B.2 IDENTIFIABILITY IN FACTORED MDP

Due to the nature of factored MDP where the causal graph is directed bipartite, each Markov equivalence class constrained under temporal precedence contains a single unique causal graph (i.e., a skeleton determines a unique causal graph since temporal precedence fully orients the edges). Given this background, it is known that the causal graph is uniquely identifiable with oracle conditional independence test (Ding et al., 2022) or score-based method (Brouillard et al., 2020). Similarly, we now show that LCG is also identifiable via score maximization.

---

[2]$T$ denotes an index set of $\mathbf{T}$.

To begin with, we recall the score function in Eq. (4):

$$\mathcal{S}(\{\mathcal{G}_z, \mathcal{E}_z\}_{z=1}^K) \tag{8}$$

$$:= \sup \mathbb{E}\left[\log \hat{p}(s' \mid s, a; \{\mathcal{G}_z, \mathcal{E}_z\}) - \lambda |\mathcal{G}_z|\right]$$

$$:= \sup_\phi \mathbb{E}_{p(s,a,s')}\left[\log \hat{p}(s' \mid s, a; \{\mathcal{G}_z, \mathcal{E}_z\}, \phi) - \lambda |\mathcal{G}_z|\right], \tag{9}$$

$$= \sup_\phi \mathbb{E}_{p(s,a)}\mathbb{E}_{p(s'|s,a)}\left[\log \hat{p}(s' \mid s, a; \{\mathcal{G}_z, \mathcal{E}_z\}, \phi) - \lambda |\mathcal{G}_z|\right], \tag{10}$$

$$= \sup_\phi \sum_z \int_{(s,a) \in \mathcal{E}_z} p(s,a)\left(\mathbb{E}_{p(s'|s,a)} \log \hat{p}(s' \mid s, a; \mathcal{G}_z, \phi) - \lambda |\mathcal{G}_z|\right), \tag{11}$$

$$= \sup_\phi \sum_z \left[\int_{(s,a) \in \mathcal{E}_z} p(s,a)\left(\mathbb{E}_{p(s'|s,a)} \log \hat{p}(s' \mid s, a; \mathcal{G}_z, \phi)\right) - \lambda \cdot p(\mathcal{E}_z) \cdot |\mathcal{G}_z|\right], \tag{12}$$

where

$$\hat{p}(s' \mid s, a; \mathcal{G}_z, \phi) = \prod_j \hat{p}_j(s'_j \mid Pa(j, z), z; \phi_j), \tag{13}$$

$\phi := \{\phi_1, \cdots, \phi_N\}$, and each $\phi_j$ is a neural network which outputs the parameters of the density function $\hat{p}_j$. Specifically, for all $z \in [K]$, $\phi_j$ takes $Pa(j, z)$ (i.e., parents of $s'_j$ in $\mathcal{G}_z$) as an input if $(s, a) \in \mathcal{E}_z$. We denote $\hat{p}_{\{\mathcal{G}_z, \mathcal{E}_z\}, \phi} := \hat{p}(s' \mid s, a; \{\mathcal{G}_z, \mathcal{E}_z\}, \phi)$ and $\hat{p}_{\mathcal{G}_z, \phi} := \hat{p}(s' \mid s, a; \mathcal{G}_z, \phi)$.

**Assumption 3** (Sufficient capacity). *The ground truth density $p(s' \mid s, a) \in \mathcal{H}(\{\mathcal{G}_z^*, \mathcal{E}_z\})$ for any decomposition $\{\mathcal{E}_z\}$ with corresponding true LCGs $\{\mathcal{G}_z^*\}$, where*

$$\mathcal{H}(\{\mathcal{G}_z^*, \mathcal{E}_z\}) := \{p \mid \exists \phi \text{ s.t. } p = \hat{p}_{\{\mathcal{G}_z^*, \mathcal{E}_z\}, \phi}\}. \tag{14}$$

*In other words, the model has sufficient capacity to represent the ground truth density. We assume the density $\hat{p}_{\{\mathcal{G}_z^*, \mathcal{E}_z\}, \phi}$ is strictly positive for all $\phi$.*

**Assumption 4** (Finite differential entropy). $|\mathbb{E}_{p(s,a,s')} \log p(s' \mid s, a)| < \infty$.

**Lemma 2** (Finite score). *Let $\mathcal{G}_z^*$ be a true LCG on $\mathcal{E}_z$ for all $z$. Then, $|\mathcal{S}(\{\mathcal{G}_z^*, \mathcal{E}_z\}_{z=1}^K)| < \infty$.*

*Proof.* First,

$$0 \le D_{KL}(p \parallel \hat{p}_{\{\mathcal{G}_z^*, \mathcal{E}_z\}, \phi}) \tag{15}$$

$$= \mathbb{E}_{p(s,a,s')} \log p(s' \mid s, a) - \mathbb{E}_{p(s,a,s')} \log \hat{p}(s' \mid s, a; \{\mathcal{G}_z^*, \mathcal{E}_z\}, \phi), \tag{16}$$

where the equality holds because $\mathbb{E}_{p(s,a,s')} \log p(s' \mid s, a) < \infty$ by Assumption 4. Therefore,

$$\sup_\phi \mathbb{E}_{p(s,a,s')} \log \hat{p}(s' \mid s, a; \{\mathcal{G}_z^*, \mathcal{E}_z\}, \phi) \le \mathbb{E}_{p(s,a,s')} \log p(s' \mid s, a). \tag{17}$$

On the other hand, by Assumption 3, there exists $\phi^*$ such that $p = \hat{p}_{\{\mathcal{G}_z^*, \mathcal{E}_z\}, \phi^*}$. Hence,

$$\sup_\phi \mathbb{E}_{p(s,a,s')} \log \hat{p}(s' \mid s, a; \{\mathcal{G}_z^*, \mathcal{E}_z\}, \phi) \ge \mathbb{E}_{p(s,a,s')} \log \hat{p}(s' \mid s, a; \{\mathcal{G}_z^*, \mathcal{E}_z\}, \phi^*) \tag{18}$$

$$= \mathbb{E}_{p(s,a,s')} \log p(s' \mid s, a). \tag{19}$$

Thus, $\sup_\phi \mathbb{E}_{p(s,a,s')} \log \hat{p}(s' \mid s, a; \{\mathcal{G}_z^*, \mathcal{E}_z\}, \phi) = \mathbb{E}_{p(s,a,s')} \log p(s' \mid s, a)$. Therefore,

$$\mathcal{S}(\{\mathcal{G}_z^*, \mathcal{E}_z\}_{z=1}^K) = \mathbb{E}_{p(s,a,s')} \log p(s' \mid s, a) - \lambda \cdot \mathbb{E}\left[|\mathcal{G}_z^*|\right]. \tag{20}$$

Since $|\mathcal{G}_z^*| \le N(N + M)$ and $|\mathbb{E}_{p(s,a,s')} \log p(s' \mid s, a)| < \infty$ by Assumption 4, this concludes that $|\mathcal{S}(\{\mathcal{G}_z^*, \mathcal{E}_z\}_{z=1}^K)| < \infty$. $\square$

**Lemma 3** (Score difference). *Let $\mathcal{G}_z^*$ be a true LCG on $\mathcal{E}_z$ for all $z$. Then,*

$$\mathcal{S}(\{\mathcal{G}_z^*, \mathcal{E}_z\}_{z=1}^K) - \mathcal{S}(\{\mathcal{G}, \mathcal{E}_z\}_{z=1}^K) = \inf_\phi D_{KL}(p \parallel \hat{p}_{\{\mathcal{G}_z, \mathcal{E}_z\}, \phi}) + \lambda \sum_z p(\mathcal{E}_z)(|\mathcal{G}_z| - |\mathcal{G}_z^*|). \tag{21}$$

*Proof.* First, we can rewrite the score $\mathcal{S}(\{\mathcal{G}_z, \mathcal{E}_z\}_{z=1}^K)$ as:

$$\mathcal{S}(\{\mathcal{G}_z, \mathcal{E}_z\}_{z=1}^K) = \sup_\phi \mathbb{E}_{p(s,a,s')} \log \hat{p}(s' \mid s, a; \{\mathcal{G}_z, \mathcal{E}_z\}, \phi) - \lambda \cdot \mathbb{E}\left[|\mathcal{G}_z|\right] \tag{22}$$

$$= -\inf_\phi -\mathbb{E}_{p(s,a,s')} \log \hat{p}(s' \mid s, a; \{\mathcal{G}_z, \mathcal{E}_z\}, \phi) - \lambda \cdot \mathbb{E}\left[|\mathcal{G}_z|\right] \tag{23}$$

$$= -\inf_\phi D_{KL}(p \parallel \hat{p}_{\{\mathcal{G}_z, \mathcal{E}_z\}, \phi}) + \mathbb{E}_{p(s,a,s')} \log p(s' \mid s, a) - \lambda \cdot \mathbb{E}\left[|\mathcal{G}_z|\right] \tag{24}$$

The last equality holds by Assumption 4. By Lemma 2, $|\mathcal{S}(\{\mathcal{G}_z^*, \mathcal{E}_z\}_{z=1}^K)| < \infty$ and thus the score difference $\mathcal{S}(\{\mathcal{G}_z^*, \mathcal{E}_z\}_{z=1}^K) - \mathcal{S}(\{\mathcal{G}_z, \mathcal{E}_z\}_{z=1}^K)$ is well defined. Using Eq. (20), we obtain:

$$\mathcal{S}(\{\mathcal{G}_z^*, \mathcal{E}_z\}_{z=1}^K) - \mathcal{S}(\{\mathcal{G}_z, \mathcal{E}_z\}_{z=1}^K) = \inf_\phi D_{KL}(p \parallel \hat{p}_{\{\mathcal{G}_z, \mathcal{E}_z\}, \phi}) + \lambda \sum_z p(\mathcal{E}_z)(|\mathcal{G}_z| - |\mathcal{G}_z^*|).$$

$\square$

**Proposition 2.** *Let $\{\mathcal{E}_z\}_{z=1}^K$ be the arbitrary decomposition. Let $\{\hat{\mathcal{G}}_z\}_{z=1}^K$ be the graphs on each $\mathcal{E}_z$ that maximizes the score: $\{\hat{\mathcal{G}}_z\}_{z=1}^K \in \mathrm{argmax}_{\{\mathcal{G}_z\}} \mathcal{S}(\{\mathcal{G}_z, \mathcal{E}_z\}_{z=1}^K)$. With the Assumptions 1 to 4, each $\hat{\mathcal{G}}_z$ is true LCG on corresponding $\mathcal{E}_z$ for small enough $\lambda > 0$. In particular, if $K = 1$, then $\hat{\mathcal{G}} = \mathcal{G}$ where $\mathcal{G}$ is the ground truth causal graph.*

*Proof.* Let $\mathcal{G}_z^*$ be a true LCG on $\mathcal{E}_z$ for all $z$. It is enough to show that $\mathcal{S}(\{\mathcal{G}_z^*, \mathcal{E}_z\}_{z=1}^K) > \mathcal{S}(\{\mathcal{G}_z, \mathcal{E}_z\}_{z=1}^K)$ if $\mathcal{G}_z^* \neq \mathcal{G}_z$ for some $z$. By Lemma 3,

$$\mathcal{S}(\{\mathcal{G}_z^*, \mathcal{E}_z\}_{z=1}^K) - \mathcal{S}(\{\mathcal{G}_z, \mathcal{E}_z\}_{z=1}^K) \tag{25}$$

$$= \inf_\phi D_{KL}(p \parallel \hat{p}_{\{\mathcal{G}_z, \mathcal{E}_z\}, \phi}) + \lambda \sum_z p(\mathcal{E}_z)(|\mathcal{G}_z| - |\mathcal{G}_z^*|) \tag{26}$$

$$= \inf_\phi \int p(s,a) D_{KL}(p(\cdot \mid s, a) \parallel \hat{p}_{\{\mathcal{G}_z, \mathcal{E}_z\}, \phi}(\cdot \mid s, a)) + \lambda \sum_z p(\mathcal{E}_z)(|\mathcal{G}_z| - |\mathcal{G}_z^*|) \tag{27}$$

$$= \inf_\phi \sum_z \int_{(s,a) \in \mathcal{E}_z} p(s,a) D_{KL}(p(\cdot \mid s, a) \parallel \hat{p}(\cdot \mid s, a; \mathcal{G}_z, \phi)) + \lambda \sum_z p(\mathcal{E}_z)(|\mathcal{G}_z| - |\mathcal{G}_z^*|) \tag{28}$$

$$= \inf_\phi \sum_z p(\mathcal{E}_z) \int p_z(s,a) D_{KL}(p(\cdot \mid s, a) \parallel \hat{p}(\cdot \mid s, a; \mathcal{G}_z, \phi)) + \lambda \sum_z p(\mathcal{E}_z)(|\mathcal{G}_z| - |\mathcal{G}_z^*|) \tag{29}$$

$$= \inf_\phi \sum_z p(\mathcal{E}_z) D_{KL}(p_z \parallel \hat{p}_{\mathcal{G}_z, \phi}) + \lambda \sum_z p(\mathcal{E}_z)(|\mathcal{G}_z| - |\mathcal{G}_z^*|) \tag{30}$$

$$= \sum_z p(\mathcal{E}_z) \inf_\phi D_{KL}(p_z \parallel \hat{p}_{\mathcal{G}_z, \phi}) + \lambda \sum_z p(\mathcal{E}_z)(|\mathcal{G}_z| - |\mathcal{G}_z^*|) \tag{31}$$

$$= \sum_z p(\mathcal{E}_z) \left[ \inf_\phi D_{KL}(p_z \parallel \hat{p}_{\mathcal{G}_z, \phi}) + \lambda(|\mathcal{G}_z| - |\mathcal{G}_z^*|) \right]. \tag{32}$$

Here, $p_z(s,a) = p(s,a)/p(\mathcal{E}_z)$, i.e., density function of the distribution $P_{S \times A \mid \mathcal{E}_z}$, i.e., state and action variables restricted to $\mathcal{E}_z$. For brevity, we denote $D_{KL}(p_z \parallel \hat{p}_{\mathcal{G}_z, \phi}) := \int p_z(s,a) D_{KL}(p(\cdot \mid s, a) \parallel \hat{p}(\cdot \mid s, a; \mathcal{G}_z, \phi))$ and $A_z := \inf_\phi D_{KL}(p_z \parallel \hat{p}_{\mathcal{G}_z, \phi}) + \lambda(|\mathcal{G}_z| - |\mathcal{G}_z^*|)$. We will show that for all $z \in [K]$, $A_z > 0$ if and only if $\mathcal{G}_z^* \neq \mathcal{G}_z$.
**Case 0:** $\mathcal{G}_z^* = \mathcal{G}_z$. Clearly, $A_z = 0$ in this case.
**Case 1:** $\mathcal{G}_z^* \subsetneq \mathcal{G}_z$. Then, $|\mathcal{G}_z| > |\mathcal{G}_z^*|$ and thus $A_z > 0$ since $\lambda > 0$.
**Case 2:** $\mathcal{G}_z^* \not\subseteq \mathcal{G}_z$. In this case, there exists $(i \to j) \in \mathcal{G}_z^*$ such that $(i \to j) \notin \mathcal{G}_z$ and thus $\inf_\phi D_{KL}(p_z \parallel \hat{p}_{\mathcal{G}_z, \phi}) > 0$. We consider two subcases: (i) $\mathcal{G}_z \in \mathbb{G}_z^+ := \{\mathcal{G}' \mid \mathcal{G}_z^* \not\subseteq \mathcal{G}', |\mathcal{G}'| \geq |\mathcal{G}_z^*|\}$, and (ii) $\mathcal{G}_z \in \mathbb{G}_z^- := \{\mathcal{G}' \mid \mathcal{G}_z^* \not\subseteq \mathcal{G}', |\mathcal{G}'| < |\mathcal{G}_z^*|\}$. Clearly, if $\mathcal{G}_z \in \mathbb{G}_z^+$ then $A_z > 0$. Suppose $\mathcal{G}_z \in \mathbb{G}_z^-$. Then,

$$\lambda \leq \eta_z := \frac{1}{N(N+M)+1} \min_{\mathcal{G}' \in \mathbb{G}_z^-} \inf_\phi D_{KL}(p_z \parallel \hat{p}_{\mathcal{G}', \phi}) \tag{33}$$

$$\Longleftrightarrow \quad \lambda \leq \frac{\inf_\phi D_{KL}(p_z \parallel \hat{p}_{\mathcal{G}', \phi})}{N(N+M)+1} < \frac{\inf_\phi D_{KL}(p_z \parallel \hat{p}_{\mathcal{G}', \phi})}{|\mathcal{G}_z^*| - |\mathcal{G}'|} \quad \text{for } \forall \mathcal{G}' \in \mathbb{G}_z^- \tag{34}$$

$$\Longleftrightarrow \quad \inf_\phi D_{KL}(p_z \parallel \hat{p}_{\mathcal{G}', \phi}) + \lambda(|\mathcal{G}_z| - |\mathcal{G}_z^*|) > 0 \quad \text{for } \forall \mathcal{G}' \in \mathbb{G}_z^-. \tag{35}$$

Here, we use the fact that $|\mathcal{G}_z^*| - |\mathcal{G}'| \leq |\mathcal{G}_z^*| < N(N+M) + 1$. Therefore, $A_z > 0$ if $\mathcal{G}_z^* \neq \mathcal{G}_z$ for $0 < \forall \lambda \leq \eta_z$. Consequently, for $0 < \lambda \leq \eta(\{\mathcal{E}_z\}) := \min_z \eta_z$, we have $\mathcal{S}(\{\mathcal{G}_z^*, \mathcal{E}_z\}_{z=1}^K) - \mathcal{S}(\{\mathcal{G}_z, \mathcal{E}_z\}_{z=1}^K) > 0$ if $\mathcal{G}_z^* \neq \mathcal{G}_z$ for some $z$. We note that (i) $\eta_z > 0$ since $\mathbb{G}_z^-$ is finite and $\inf_\phi D_{KL}(p_z \parallel \hat{p}_{\mathcal{G}',\phi}) > 0$ for any $\mathcal{G}' \in \mathbb{G}_z^-$, and thus (ii) $\eta(\{\mathcal{E}_z\}) = \min_z \eta_z > 0$. $\square$

**Assumption 5.** $\inf_{\{\mathcal{E}_z\} \in \mathcal{T}} \eta(\{\mathcal{E}_z\}) > 0$ where $\mathcal{T}$ is a set of all decompositions of size $K$.

Recall that Prop. 2 holds for $0 < \lambda \leq \eta(\{\mathcal{E}_z\})$ and $\eta(\{\mathcal{E}_z\}) > 0$ for any decomposition $\{\mathcal{E}_z\}$. With Assumption 5, we now let $0 < \lambda \leq \inf_{\{\mathcal{E}_z\} \in \mathcal{T}} \eta(\{\mathcal{E}_z\})$, which allows Prop. 2 to hold on any arbitrary decomposition. It is worth noting that $\mathcal{T}$ is a highly complex mathematical object, which makes it challenging to prove or find a counterexample of the above assumption. In general, for a small fixed $\lambda > 0$, the arguments henceforth would hold for decompositions $\{\mathcal{E}_z\} \in \mathcal{T}_\lambda = \{\{\mathcal{E}_z\}_{z=1}^K \mid \eta(\{\mathcal{E}_z\}) \geq \lambda\}$, where $\mathcal{T}_\lambda \to \mathcal{T}$ as $\lambda \to 0$.

**Proposition 3.** *Let $\{\hat{\mathcal{G}}_z, \hat{\mathcal{E}}_z\}_{z=1}^K \in \arg\max_{\{\mathcal{G}_z, \mathcal{E}_z\}} \mathcal{S}(\{\mathcal{G}_z, \mathcal{E}_z\}_{z=1}^K)$. Then, each $\hat{\mathcal{G}}_z$ is the true LCG on $\hat{\mathcal{E}}_z$ for all $z \in [K]$. Also, let $\{\mathcal{F}_z\}_{z=1}^K$ be the arbitrary decomposition and $\{\mathcal{G}_z\}_{z=1}^K$ be the corresponding true LCGs on each $\mathcal{F}_z$. Then, with the Assumptions 1 to 5, $\mathbb{E}\left[|\hat{\mathcal{G}}_z|\right] \leq \mathbb{E}\left[|\mathcal{G}_z|\right]$ holds for small enough $\lambda > 0$.*

*Proof.* Let $0 < \lambda \leq \inf_{\{\mathcal{E}_z\} \in \mathcal{T}} \eta(\{\mathcal{E}_z\})$. First, $\{\hat{\mathcal{G}}_z, \hat{\mathcal{E}}_z\}_{z=1}^K \in \arg\max_{\{\mathcal{G}_z, \mathcal{E}_z\}} \mathcal{S}(\{\mathcal{G}_z, \mathcal{E}_z\}_{z=1}^K)$ implies that $\{\hat{\mathcal{G}}_z\}_{z=1}^K$ also maximizes the score on the fixed $\{\hat{\mathcal{E}}_z\}_{z=1}^K$, i.e., $\{\hat{\mathcal{G}}_z\}_{z=1}^K \in \arg\max_{\{\mathcal{G}_z\}} \mathcal{S}(\{\mathcal{G}_z, \hat{\mathcal{E}}_z\}_{z=1}^K)$. Thus, $\hat{\mathcal{G}}_z$ is true LCG on $\mathcal{E}_z$ for all $z \in [K]$ by Prop. 2. Also, since $\{\mathcal{F}_z\}_{z=1}^K$ is the arbitrary decomposition, $\mathcal{S}(\{\hat{\mathcal{G}}_z, \hat{\mathcal{E}}_z\}) \geq \mathcal{S}(\{\mathcal{G}_z, \mathcal{F}_z\})$ holds. Since $\{\mathcal{G}_z\}$ is the true LCGs on each $\mathcal{F}_z$, by Eq. (20),

$$\mathcal{S}(\{\mathcal{G}_z, \mathcal{F}_z\}_{z=1}^K) = \mathbb{E}_{p(s,a,s')} \log p(s' \mid s, a) - \lambda \sum_z p(\mathcal{F}_z) \cdot |\mathcal{G}_z|. \tag{36}$$

Similarly,

$$\mathcal{S}(\{\hat{\mathcal{G}}_z, \hat{\mathcal{E}}_z\}_{z=1}^K) = \mathbb{E}_{p(s,a,s')} \log p(s' \mid s, a) - \lambda \sum_z p(\hat{\mathcal{E}}_z) \cdot |\hat{\mathcal{G}}_z|. \tag{37}$$

Therefore, $0 \leq \mathcal{S}(\{\hat{\mathcal{G}}_z, \hat{\mathcal{E}}_z\}) - \mathcal{S}(\{\mathcal{G}_z, \mathcal{F}_z\}) = \mathbb{E}\left[|\mathcal{G}_z|\right] - \mathbb{E}\left[|\hat{\mathcal{G}}_z|\right]$ holds. $\square$

We note that Prop. 3 can be further generalized because a partition of size $K$ can express any partition of size $J \leq K$. For example, the partition $\{\mathcal{E}_k\}_{k=1}^K$ can express the partition $\{\mathcal{D}_j\}_{j=1}^J$ by letting $\mathcal{E}_1 = \mathcal{D}_1, \cdots, \mathcal{E}_{J-1} = \mathcal{D}_{J-1}$, and $\{\mathcal{E}_J, \cdots, \mathcal{E}_K\}$ be a partition of $\mathcal{D}_J$.

**Theorem 1** (Identifiability). *Let $\{\hat{\mathcal{G}}_z, \hat{\mathcal{E}}_z\}_{z=1}^K \in \arg\max_{\{\mathcal{G}_z, \mathcal{E}_z\}} \mathcal{S}(\{\mathcal{G}_z, \mathcal{E}_z\}_{z=1}^K)$ and $K > 1$. Let $\mathcal{D} \subsetneq \mathcal{X}$ where $\mathcal{G}_\mathcal{D} \subsetneq \mathcal{G}$. Suppose $\mathcal{G}_\mathcal{F} = \mathcal{G}_\mathcal{D}$ for any $\mathcal{F} \subseteq \mathcal{D}$, and $\mathcal{G}_\mathcal{F} = \mathcal{G}$ for any $\mathcal{F} \subseteq \mathcal{D}^c$. Then, with the Assumptions 1 to 5, there exists $I \subsetneq [K]$ such that (i) $\bigcup_{i \in I} \hat{\mathcal{E}}_i \subseteq \mathcal{D}$ and $p(\mathcal{D} \setminus \bigcup_{i \in I} \hat{\mathcal{E}}_i) = 0$, (ii) $\hat{\mathcal{G}}_z = \mathcal{G}_\mathcal{D}$ for all $z \in I$ and $\hat{\mathcal{G}}_z = \mathcal{G}$ for all $z \notin I$.*

*Proof.* For brevity, we denote the conditions $\mathcal{G}_\mathcal{F} = \mathcal{G}_\mathcal{D}$ for any $\mathcal{F} \subseteq \mathcal{D}$ as condition (i) , and $\mathcal{G}_\mathcal{F} = \mathcal{G}$ for any $\mathcal{F} \subseteq \mathcal{D}^c$ as condition (ii). Let $\{\mathcal{F}_z\}_{z=1}^K$ be the decomposition such that $\bigcup_{j \in J} \mathcal{F}_j = \mathcal{D}$ for some $J \subsetneq [K]$. Let $\{\mathcal{G}_z\}_{z=1}^K$ be the LCGs corresponding to each $\mathcal{F}_z$. For any $z \in J$, $\mathcal{F}_z \subseteq \mathcal{D}$ and thus $\mathcal{G}_z = \mathcal{G}_\mathcal{D}$ by Prop. 1 and condition (i). For any $z \notin J$, $\mathcal{F}_z \subseteq \mathcal{D}^c$ and thus $\mathcal{G}_\mathcal{F} = \mathcal{G}$ by condition (ii) Therefore,

$$\mathbb{E}\left[|\mathcal{G}_z|\right] = \sum_z p(\mathcal{F}_z)|\mathcal{G}_z| = p(\mathcal{D})|\mathcal{G}_\mathcal{D}| + p(\mathcal{D}^c)|\mathcal{G}| \tag{38}$$

holds. Similarly, we let $I \in [K]$ such that $I = \{z \mid \hat{\mathcal{E}}_z \subseteq \mathcal{D}\}$. Then, $\hat{\mathcal{G}}_z = \mathcal{G}_\mathcal{D}$ for all $z \in I$ by Prop. 1 and condition (i). Also, for all $z \notin I$, let $\mathcal{T}_z := \hat{\mathcal{E}}_z \setminus \mathcal{D} \neq \emptyset$. Then, $\mathcal{T}_z \subseteq \mathcal{D}^c$ and thus $\mathcal{G}_{\mathcal{T}_z} = \mathcal{G}_{\mathcal{D}^c}$ by condition (ii). Therefore, $\mathcal{G}_{\mathcal{T}_z} = \mathcal{G}$. Since $\mathcal{T}_z \subseteq \hat{\mathcal{E}}_z$, $\mathcal{G}_{\mathcal{T}_z} \subseteq \hat{\mathcal{G}}_z$ by Prop. 1. Therefore, $\hat{\mathcal{G}}_z = \mathcal{G}$ for all $z \notin I$. Combining together,

$$\mathbb{E}\left[|\hat{\mathcal{G}}_z|\right] = \sum_z p(\hat{\mathcal{E}}_z)|\hat{\mathcal{G}}_z| = p(\bigcup_{i \in I} \hat{\mathcal{E}}_i)|\mathcal{G}_\mathcal{D}| + p(\bigcup_{i \notin I} \hat{\mathcal{E}}_i)|\mathcal{G}| \tag{39}$$

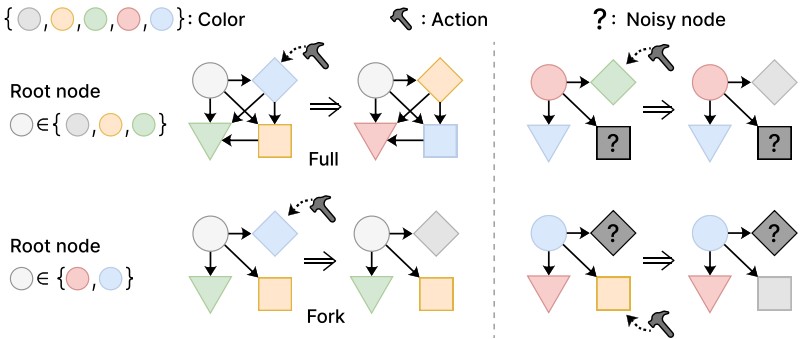

Figure 6: Illustration of CHEMICAL (*full-fork*) environment with 4 nodes. (**Left**) the color of the root node determines the activation of local causal graph *fork*. (**Right**) the noisy nodes are redundant for predicting the colors of other nodes under the local causal graph.

holds. Recall that $\mathbb{E}\left[|\hat{\mathcal{G}}_z|\right] \le \mathbb{E}\left[|\mathcal{G}_z|\right]$ holds by Prop. 3. Also, by definition of $I$, $\bigcup_{i \in I} \hat{\mathcal{E}}_i \subseteq \mathcal{D}$. By subtracting Eq. (39) from Eq. (38),

$$p(\mathcal{D} \setminus \bigcup_{i \in I} \hat{\mathcal{E}}_i) \cdot (|\mathcal{G}_{\mathcal{D}}| - |\mathcal{G}|) \ge 0 \tag{40}$$

holds. Since $|\mathcal{G}_{\mathcal{D}}| < |\mathcal{G}|$ , this is only possible when $p(\mathcal{D} \setminus \bigcup_{i \in I} \hat{\mathcal{E}}_i) = 0$. $\qquad\square$

## C  APPENDIX FOR EXPERIMENTS

### C.1  EXPERIMENTAL DETAILS

#### C.1.1  PLANNING ALGORITHM

To assess the performance of different dynamics models of the baselines and our method, we use a model predictive control (MPC) (Camacho & Alba, 2013) which selects the actions based on the prediction of the learned dynamics model, following prior works (Ding et al., 2022; Wang et al., 2022). Specifically, we use a cross-entropy method (CEM) (Rubinstein & Kroese, 2004), which iteratively generates and refines action sequences through a process of sampling from a probability distribution that is updated based on the performance of these sampled sequences, with a known reward function. We use a random policy for the initial data collection.

Table 4: Environment configurations.

| Parameters | Chemical | | Magnetic |
| --- | --- | --- | --- |
| | *full-fork* | *full-chain* | |
| Training step | $1.5 \times 10^5$ | $1.5 \times 10^5$ | $2 \times 10^5$ |
| Optimizer | Adam | Adam | Adam |
| Learning rate | 1e-4 | 1e-4 | 1e-4 |
| Batch size | 256 | 256 | 256 |
| Initial step | 1000 | 1000 | 2000 |
| Max episode length | 25 | 25 | 25 |
| Action type | Discrete | Discrete | Continuous |

Table 5: CEM parameters.

| CEM parameters | Chemical | | Magnetic |
| --- | --- | --- | --- |
| | *full-fork* | *full-chain* | |
| Planning length | 3 | 3 | 1 |
| Number of candidates | 64 | 64 | 64 |
| Number of top candidates | 32 | 32 | 32 |
| Number of iterations | 5 | 5 | 5 |
| Exploration noise | N/A | N/A | 1e-4 |
| Exploration probability | 0.05 | 0.05 | N/A |

#### C.1.2  CHEMICAL

Here, we describe two settings, namely *full-fork* and *full-chain*, modified from Ke et al. (2021). In both settings, there are 10 state variables representing the color of corresponding nodes, with each color represented as a one-hot encoding. The action variable is a 50-dimensional categorical variable that changes the color of a specific node to a new color (e.g., changing the color of the third node to blue). According to the underlying causal graph and pre-defined conditional probability distributions, implemented with randomly initialized neural networks, an action changes the colors of

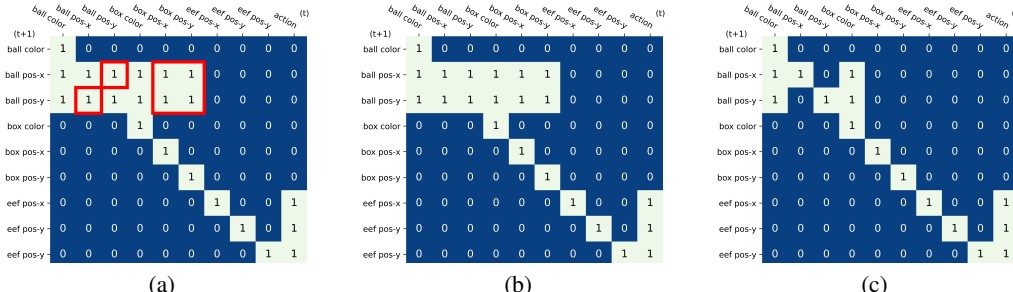

Figure 7: (a) Causal graph of Magnetic environment. Red boxes indicate redundant edges under the non-magnetic context. (b) LCG under the magnetic context. (c) LCG under the non-magnetic context.

the intervened object's descendants as depicted in Fig. 6. As shown in Fig. 3(a), the (global) causal graph is *full* in both settings, and the LCG is *fork* and *chain*, respectively. For example in *full-fork*, the LCG *fork* is activated according to the particular color of the root node, as shown in Fig. 6.

In both settings, the task is to match the colors of each node to the given target. The reward function is defined as:

$$r = \frac{1}{|O|} \sum_{i \in O} \mathbb{1}\left[s_i = g_i\right], \tag{41}$$

where $O$ is a set of the indices of observable nodes, $s_i$ is the current color of the $i$-th node, and $g_i$ is the target color of the $i$-th node in this episode. Success is determined if all colors of observable nodes are the same as the target. During training, all 10 nodes are observable, i.e., $O = \{0, \cdots, 9\}$. In downstream tasks, the root color is set to induce the LCG, and the agent receives noisy observations for a subset of nodes, aiming to match the colors of the rest of the observable nodes. As shown in Fig. 6, noisy nodes are spurious for predicting the colors of other nodes under the LCG. Thus, the agent capable of reasoning the fine-grained causal relationships would generalize well in downstream tasks.[3] To create noisy observations, we use a noise sampled from $\mathcal{N}(0, \sigma^2)$, similar to Wang et al. (2022). Specifically, the noise is multiplied to the one-hot encoding representing color during the test. In our experiments, we use $\sigma = 100$.

As the root color determines the local causal graph in both settings, the root node is always observable to the agent during the test. The root colors of the initial state and the goal state are the same, inducing the local causal graph. As the root color can be changed by the action during the test, this may pose a challenge in evaluating the agent's reasoning of local causal relationships. This can be addressed by modifying the initial distribution of CEM to exclude the action on the root node and only act on the other nodes during the test. Nevertheless, we observe that restricting the action on the root during the test has little impact on the behavior of any model, and we find that this is because the agent rarely changes the root color as it already matches the goal color in the initial state.

### C.1.3 MAGNETIC

In this environment, there are two objects on a table, a moving ball and a box, colored either red or black, as shown in Fig. 3(b). The red color indicates that the object is *magnetic*. In other words, when they are both colored red, magnetic force will be applied and the ball will move toward the box. If one of the objects is colored black, the ball would not move since the box has no influence on the ball. The state consists of the color, $x, y$ position of each object, and $x, y, z$ position of the end-effector of the robot arm, where the color is given as the 2-dimensional one-hot encoding. The action is a 3-dimensional vector that moves the robot arm. The causal graph of the Magnetic environment is shown in Fig. 7(a). LCGs under magnetic and non-magnetic event are shown in Figs. 7(b) and 7(c), respectively. The table in our setup has a width of 0.9 and a length of 0.6, with the y-axis defined by the width and the x-axis defined by the length. For each episode, the initial positions of a moving ball and a box are randomly sampled within the range of the table.

---

[3]We note that the transition dynamics of the environment is the same in training and downstream tasks.

The task is to move the robot arm to reach the moving ball. Thus, accurately predicting the trajectory of the ball is crucial. The reward function is defined as:

$$r = 1 - \tanh(5 \cdot \|eef - g\|_1), \tag{42}$$

where the $eef \in \mathbb{R}^3$ is the current position of the end-effector, $g = (b_x, b_y, 0.8) \in \mathbb{R}^3$, and $(b_x, b_y)$ is the current position of the moving ball. Success is determined if the distance is smaller than 0.05. During the test, the color of one of the objects is black and the box is located at the position unseen during the training. Specifically, the box position is sampled from $\mathcal{N}(0, \sigma^2)$ during the test. Note that the box can be located outside of the table, which never happens during the training. In our experiments, we use $\sigma = 100$.

## C.2 IMPLEMENTATION DETAILS

For all methods, the dynamics model outputs the parameters of categorical distribution for discrete variables, and the mean and standard deviation of normal distribution for continuous variables. Each method has a similar number of model parameters. All experiments were processed using a single GPU (NVIDIA RTX 3090). Environmental configurations and CEM parameters are shown in Table 4 and Table 5, respectively. Detailed parameters of each model are shown in Table 6.

**MLP and Modular.** MLP models the transition dynamics as $p(s' \mid s, a)$. Modular has a separate network for each state variable, i.e., $\prod_j p(s'_j \mid s, a)$, where each network is implemented as an MLP.

**GNN, NPS, and CDL.** We employ publicly available source codes.[4] For NPS, we search the number of rules $N \in \{4, 15, 20\}$. CDL infers the causal structure utilizing conditional mutual information and models the dynamics as $\prod_j p(s'_j \mid Pa(j))$. For CDL, we search the initial conditional mutual information (CMI) threshold $\epsilon \in \{0.001, 0.002, 0.005, 0.01, 0.02\}$ and exponential moving average (EMA) coefficient $\tau \in \{0.9, 0.95, 0.99, 0.999\}$. As CDL is a two-stage method, we only report their final performance.

**GRADER.** We implement GRADER based on the code provided by the authors.[5] GRADER relies on explicit conditional independence testing to discover the causal structure. In Chemical, we ran the conditional independence test for every 10 episodes, following their default setting. We only report their performance in Chemical due to its poor scalability on the conditional independence test in Magnetic environment, which took about 30 minutes for each test.

**Oracle and NCD.** For a fair comparison, we employ the same architecture for the dynamic models of Oracle, NCD, and our method, as their main difference lies in the inference of local causal graphs (LCG). As illustrated in Fig. 8, the key difference is that NCD performs direct inference of the LCG from each individual sample (referred to as *sample-specific* inference), while our method decomposes the data domain and infers the LCGs for each event (referred to as *event-specific* inference). We provide an implementation details of our method in the next subsection.

### C.2.1 IMPLEMENTATION DETAILS OF OUR METHOD.

We use MLPs for the implementation of $g_{\text{enc}}, g_{\text{dec}}$, and $\hat{p}$, with configurations provided in Table 6. The quantization encoder $g_{\text{enc}}$ of our method or the auxiliary network of NCD shares the initial feature extraction layer with the dynamics model $\hat{p}$ as we found that it yields better performance compared to full decoupling of them.

**Masked dynamics model.** For the implementation of $\hat{p}(s'_j \mid s, a; A_j)$, we simply mask out the features of unused variables, but other design choices such as Gated Recurrent Unit (Chung et al., 2014; Ding et al., 2022) are also possible. As architectural design is not the primary focus of our work, we leave the exploration of different architectures to future work. Recall Eq. (3) that $p(s'_j \mid s, a) = p(s'_j \mid Pa(j; \mathcal{E}_z), z)$ for $(s, a) \in \mathcal{E}_z$, the dynamics prediction model takes not only $Pa(j; \mathcal{E}_z)$, but also $z$ as an input. This is because the transition function could be different among partitions with the same LCG in general. Here, $z$ guides the network to learn (possibly) different transition functions even if the LCG is the same. Recall that each latent code $e_z \in C = \{e_z\}_{z=1}^K$

---

[4] https://github.com/wangzizhao/CausalDynamicsLearning
[5] https://github.com/GilgameshD/GRADER

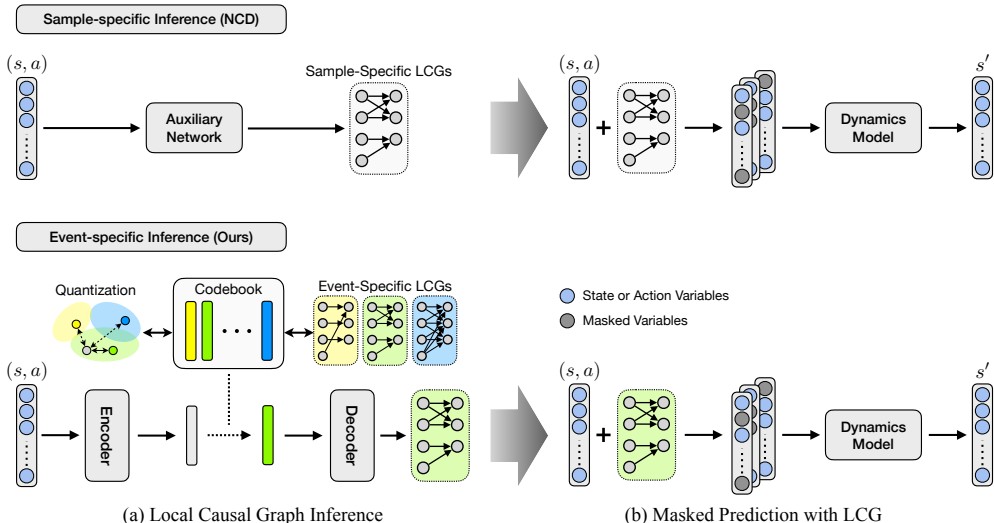

(a) Local Causal Graph Inference      (b) Masked Prediction with LCG

Figure 8: Comparison of the sample-specific inference of NCD (Hwang et al., 2023) **(top)** and event-specific inference of our method **(bottom)**.

denotes the partition, $\hat{p}$ takes a one-hot encoding of size $K$ according to the latent code as the additional input to deal with such cases.

**Backpropagation.** We now describe how each component of our method are updated by the training objective $\mathcal{L}_{\texttt{total}} = \mathcal{L}_{\texttt{pred}} + \mathcal{L}_{\texttt{quant}}$.

- In Eq. (6), $\mathcal{L}_{\texttt{pred}}$ updates the encoder $g_{\texttt{enc}}(s,a)$, decoder $g_{\texttt{dec}}(e)$, and the dynamics model $\hat{p}$. Recall that $A \sim g_{\texttt{dec}}(e)$, backpropagation from $A$ in $\mathcal{L}_{\texttt{pred}}$ updates the quantization decoder $g_{\texttt{dec}}$ through $e$. During the backward path in Eq. (5), we copy gradients from $e$ (= input of $g_{\texttt{dec}}$) to $h$ (= output of $g_{\texttt{enc}}$), following a popular trick used in VQ-VAE (Van Den Oord et al., 2017). By doing so, $\mathcal{L}_{\texttt{pred}}$ also updates the quantization encoder $g_{\texttt{enc}}$ and $h$.

- In Eq. (7), $\mathcal{L}_{\texttt{quant}}$ updates $g_{\texttt{enc}}$ and the codebook $C$. We note that $\mathcal{L}_{\texttt{pred}}$ also affects the learning of the codebook $C$ since $h$ is updated with $\mathcal{L}_{\texttt{pred}}$.

**Hyperparameters.** For all experiments, we fix the codebook size $K = 16$, regularization coefficient $\lambda = 0.001$, and commitment coefficient $\beta = 0.25$, as we found that the performance did not vary much for any $K > 2$, $\lambda \in \{10^{-4}, 10^{-3}, 10^{-2}\}$ and $\beta \in \{0.1, 0.25\}$.

## C.3 ADDITIONAL EXPERIMENTAL RESULTS AND DISCUSSIONS

### C.3.1 DETAILED ANALYSIS OF LEARNED LCGs

LCGs learned by our method with a codebook size of 4 in Chemical are shown in Figs. 9 and 10. Among the 4 codes, one (Fig. 9(b)) or two (Fig. 10(b)) represent the local causal structure *fork*. Our method successfully infers the proper code for most of the OOD samples (Figs. 9(c) and 10(c)). Two sample runs of our method with a codebook size of 4 in Magnetic are shown in Figs. 11 and 12. Our method successfully learns LCGs correspond to a non-magnetic event (Figs. 11(d), 11(g), 12(d) and 12(f)) and magnetic event (Figs. 11(e), 11(f), 12(e) and 12(g)).

We also observe that our method discovers more fine-grained events. Recall that the non-magnetic event is determined when one of the objects is black, the box would have no influence on the ball regardless of the color of the box when the ball is black, and vice versa. As shown in Fig. 13, our method discovers the event where the ball is black (Fig. 13(b)), and the event where the box is black (Fig. 13(a)).

We observe that the training of latent codebook with vector quantization is often unstable when $K = 2$. We demonstrate the success (Fig. 14) and failure (Fig. 15) cases of our method with a codebook size of 2. In a failure case, we observe that the embeddings frequently fluctuate between

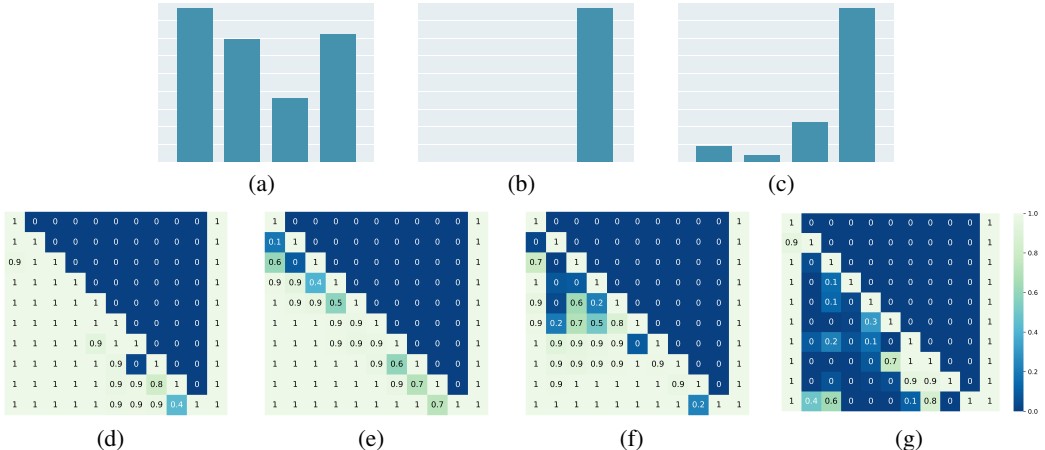

Figure 9: Analysis of LCGs learned by our method with a codebook size of 4 in Chemical (*full-fork*) environment. (a-c) Codebook histogram on (a) ID states, (b) ID states on local structure *fork*, and (c) OOD states on local structure. (d-g) Learned LCGs. The descriptions of the histograms are also applied to Figs. 10 to 12, 14 and 15.

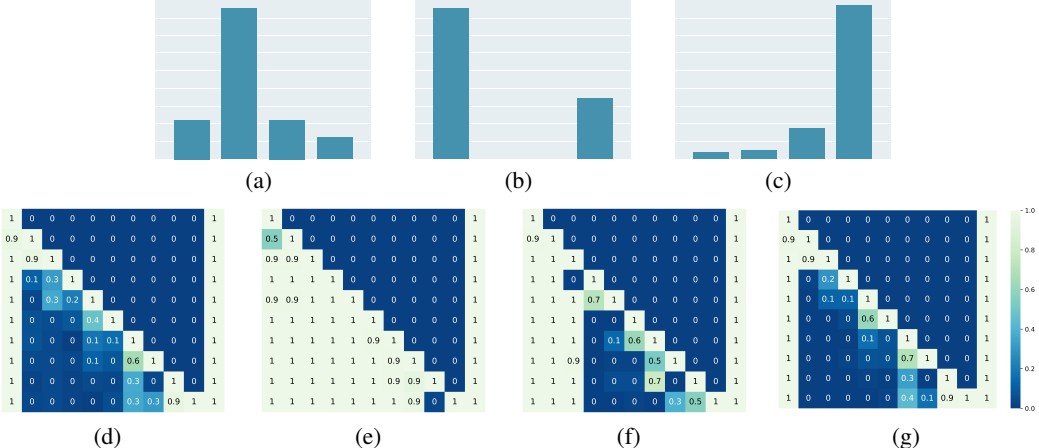

Figure 10: Another sample run of our method with a codebook size of 4 in Chemical (*full-fork*).

the two codes, resulting in both codes corresponding to the global causal graph and failing to capture the LCG, as shown in Fig. 15.

### C.3.2 EVALUATION OF LOCAL CAUSAL DISCOVERY

The performance of our method and NCD in local causal discovery is evaluated using the Structural Hamming Distance (SHD) in Magnetic. Structural Hamming Distance (SHD) is a metric used to quantify the dissimilarity between two graphs based on the number of edge additions or deletions needed to make the graphs identical (Acid & de Campos, 2003; Ramsey et al., 2006). As shown in Fig. 16, our method consistently outperforms NCD across various codebook sizes except for $K = 1$, where our method learns only a single causal graph over the entire data domain. In Fig. 16, SHD scores are averaged over the data samples in the evaluation batch. For the samples in magnetic context (i.e., both objects are red), we compare the inferred LCG with the global causal graph to measure SHD. For the samples in non-magnetic context (i.e., one of the objects is black), we compare with the one without redundant edges indicated with red boxes, as shown in Fig. 7(a). For example, as shown in Fig. 5(a), our method with $K = 1$ infers a (global) causal graph correctly and shows the SHD score of 6 in non-magnetic samples (Fig. 16, center) since inferred (global) causal graph includes redundant edges in non-magnetic events (i.e., red boxes in Fig. 7(a)).

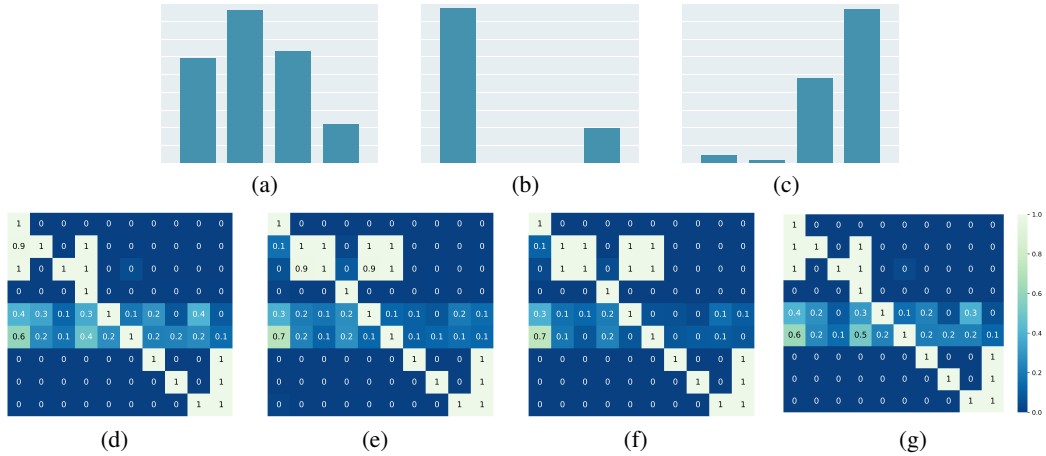

Figure 11: Analysis of LCGs learned by our method with a codebook size of 4 in Magnetic.

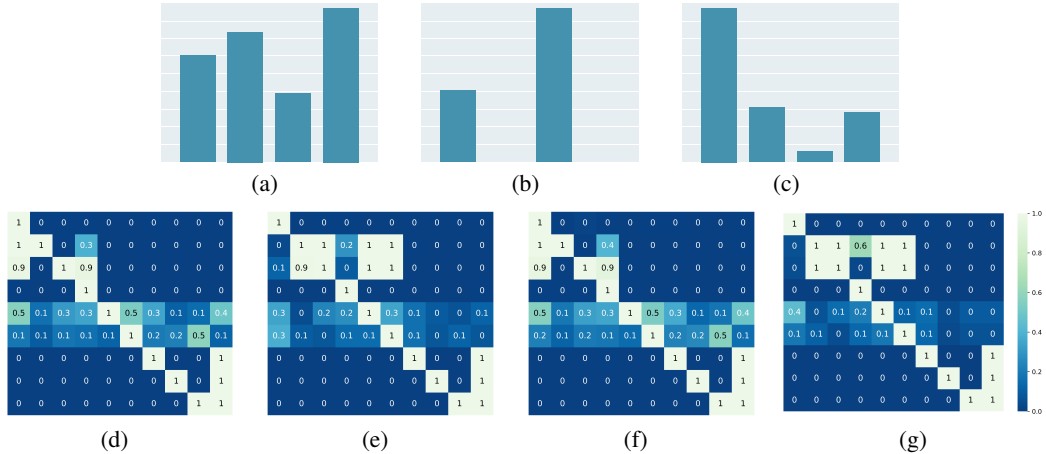

Figure 12: Another sample run of our method with a codebook size of 4 in Magnetic.

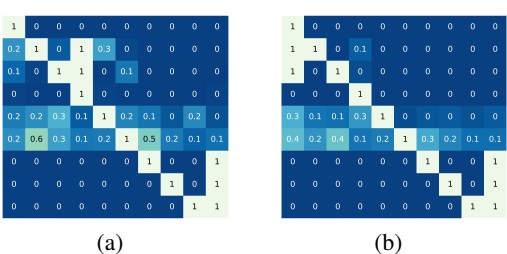

Figure 13: More fine-grained LCGs learned by our method with a codebook size of 16 in Magnetic.

### C.3.3 ADDITIONAL DISCUSSIONS

**Training with vector quantization.** It is known that training discrete latent codebook with vector quantization often suffers from the codebook collapsing, a well known issue in VQ-VAE literature. As discussed in Appendix C.3.1, we observe similar behavior when training our method when $K = 2$. Techniques have been recently proposed to prevent collapsing, such as codebook reset (Williams et al., 2020) and stochastic quantization (Takida et al., 2022). We consider that incorporating such techniques and tricks to further stabilize the training would be a future direction. We note that prior works on learning discrete latent codebook mostly focused on the reconstruction of the observation

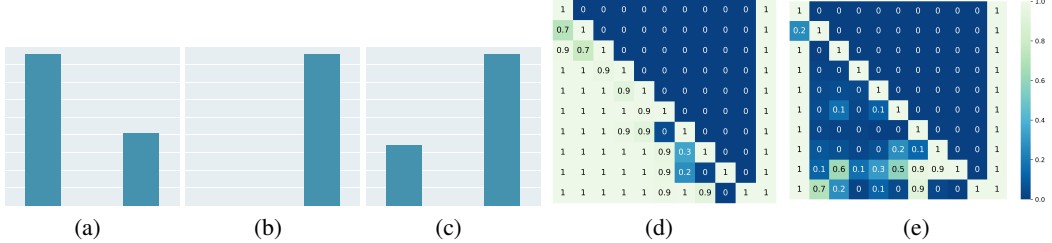

Figure 14: Analysis of LCGs learned by our method with a codebook size of 2 in Chemical (*full-fork*).

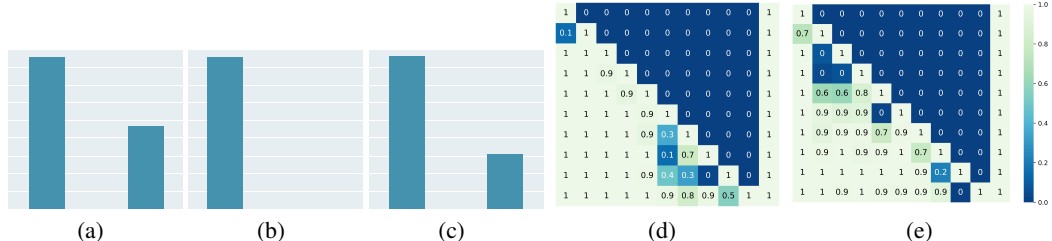

Figure 15: Failure case of our method with a codebook size of 2 in Chemical (*full-fork*).

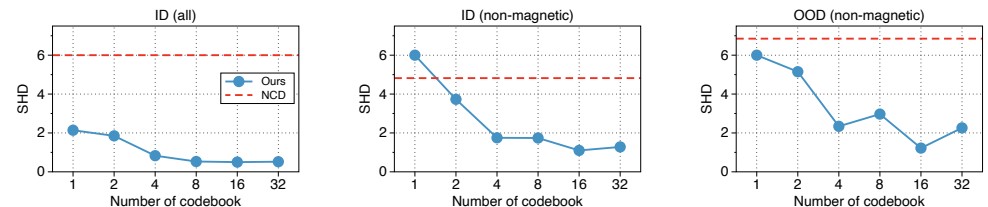

Figure 16: Evaluation of local causal discovery of NCD and our method in Magnetic environment.

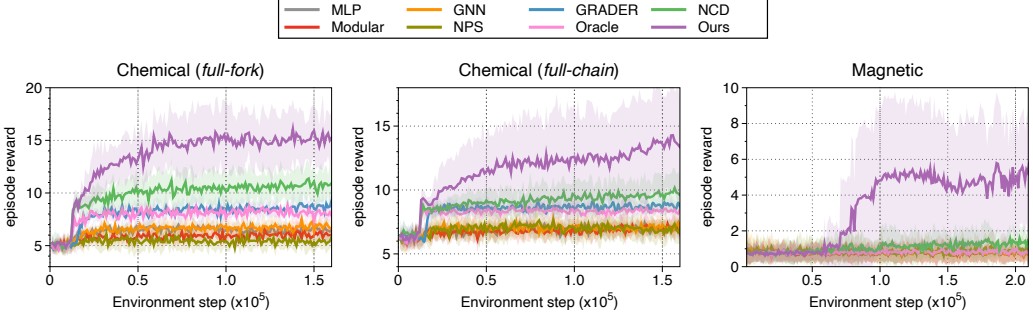

Figure 17: Learning curves on downstream tasks as measured on the average episode reward. Lines and shaded areas represent the mean and standard deviation, respectively.

(Van Den Oord et al., 2017; Ozair et al., 2021) where the size of the codebook is much larger (e.g., 128, 512, or 1024) and the utilization of vector quantization is quite different from ours.

**Relationship with Hwang et al. (2023).** Our work draws inspiration from Hwang et al. (2023), which first discussed event-level decomposition. However, NCD, their proposed method, does not explicitly discover such partitions but only infers LCG for each sample, as depicted in Fig. 8. In contrast, our method explicitly discovers $\{\mathcal{G}_z, \mathcal{E}_z\}$, decomposition and corresponding LCGs. By explicitly clustering samples into events and learning LCGs over each event, our method is more robust on OOD states than sample-specific inference. When $K = 1$, our method is equivalent to the score-based (global) causal discovery methods (Wang et al., 2021; Brouillard et al., 2020). As $K \to \infty$, our method recovers NCD, a sample-specific inference method.

Table 6: Parameters of each model.

| Models | Parameters | Chemical | | Magnetic |
|---|---|---|---|---|
| | | full-fork | full-chain | |
| MLP | Hidden dim | 1024 | 1024 | 512 |
| | Hidden layers | 3 | 3 | 4 |
| Modular | Hidden dim | 128 | 128 | 128 |
| | Hidden layers | 4 | 4 | 4 |
| GNN | Node attribute dim | 256 | 256 | 256 |
| | Node network hidden dim | 512 | 512 | 512 |
| | Node network hidden layers | 3 | 3 | 3 |
| | Edge attribute dim | 256 | 256 | 256 |
| | Edge network hidden dim | 512 | 512 | 512 |
| | Edge network hidden layers | 3 | 3 | 3 |
| NPS | Number of rules | 20 | 20 | 15 |
| | Cond selector dim | 128 | 128 | 128 |
| | Rule embedding dim | 128 | 128 | 128 |
| | Rule selector dim | 128 | 128 | 128 |
| | Feature encoder hidden dim | 128 | 128 | 128 |
| | Feature encoder hidden layers | 2 | 2 | 2 |
| | Rule network hidden dim | 128 | 128 | 128 |
| | Rule network hidden layers | 3 | 3 | 3 |
| CDL | Hidden dim | 128 | 128 | 128 |
| | Hidden layers | 4 | 4 | 4 |
| | CMI threshold | 0.001 | 0.001 | 0.001 |
| | CMI optimization frequency | 10 | 10 | 10 |
| | CMI evaluation frequency | 10 | 10 | 10 |
| | CMI evaluation step size | 1 | 1 | 1 |
| | CMI evaluation batch size | 256 | 256 | 256 |
| | EMA discount | 0.9 | 0.9 | 0.99 |
| Grader | Feature embedding dim | 128 | 128 | N/A |
| | GRU hidden dim | 128 | 128 | N/A |
| | Causal discovery frequency | 10 | 10 | N/A |
| Oracle | Hidden dim | 128 | 128 | 128 |
| | Hidden layers | 4 | 4 | 5 |
| NCD | Hidden dim | 128 | 128 | 128 |
| | Hidden layers | 4 | 4 | 5 |
| | Auxiliary network hidden dim | 128 | 128 | 128 |
| | Auxiliary network hidden layers | 2 | 2 | 2 |
| Ours | Hidden dim | 128 | 128 | 128 |
| | Hidden layers | 4 | 4 | 5 |
| | VQ encoder | [128, 64] | [128, 64] | [128, 64] |
| | VQ decoder | [32] | [32] | [32] |
| | Codebook size | 16 | 16 | 16 |
| | Code dimension | 16 | 16 | 16 |

### C.3.4 LEARNING CURVES ON ALL DOWNSTREAM TASKS

Fig. 18 shows the learning curves on training in all environments. Figs. 17, 19 and 20 shows the learning curves on all downstream tasks.[6]

### C.3.5 LIMITATIONS AND FUTURE WORK.

Insufficient or biased data may lead to inaccurate learning of causal relationships. While we assumed causal sufficiency, external factors or unobserved variables may also influence the causal relationships. Future research directions include combining our method with explicit conditional independence testing, and extending our framework to high-dimensional observation such as image, where representation learning is crucial (Schölkopf et al., 2021).

---

[6]As CDL is a two-stage method that requires searching the best threshold after the first stage training, we only report their final performance.

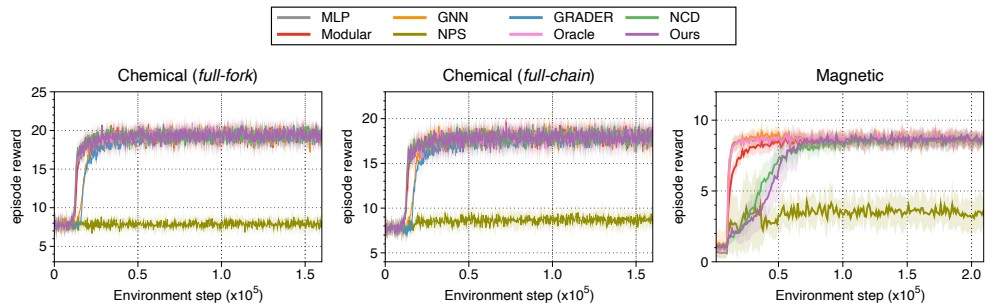

Figure 18: Learning curves during training as measured by the episode reward.

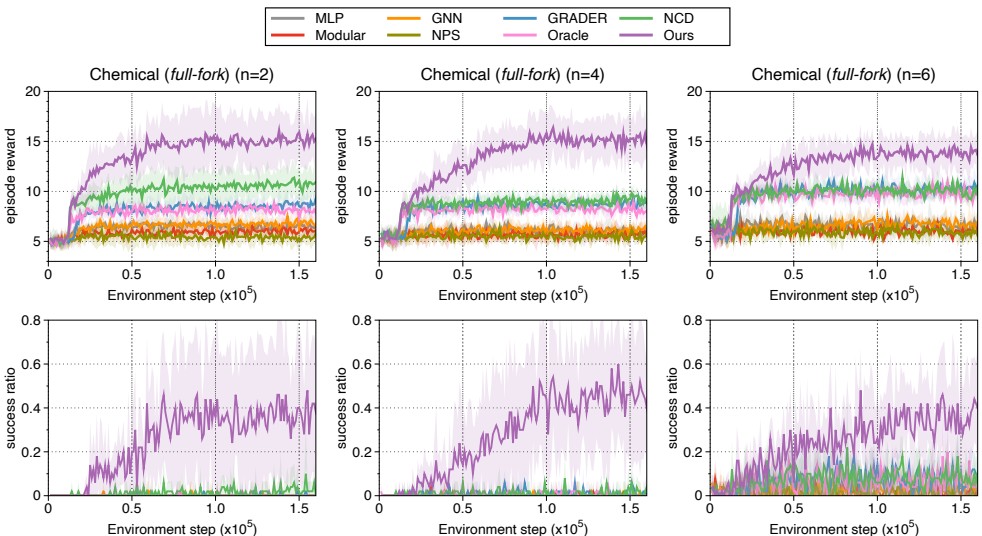

Figure 19: Learning curves on downstream tasks in Chemical (*full-fork*) as measured on the episode reward (**top**) and success rate (**bottom**).

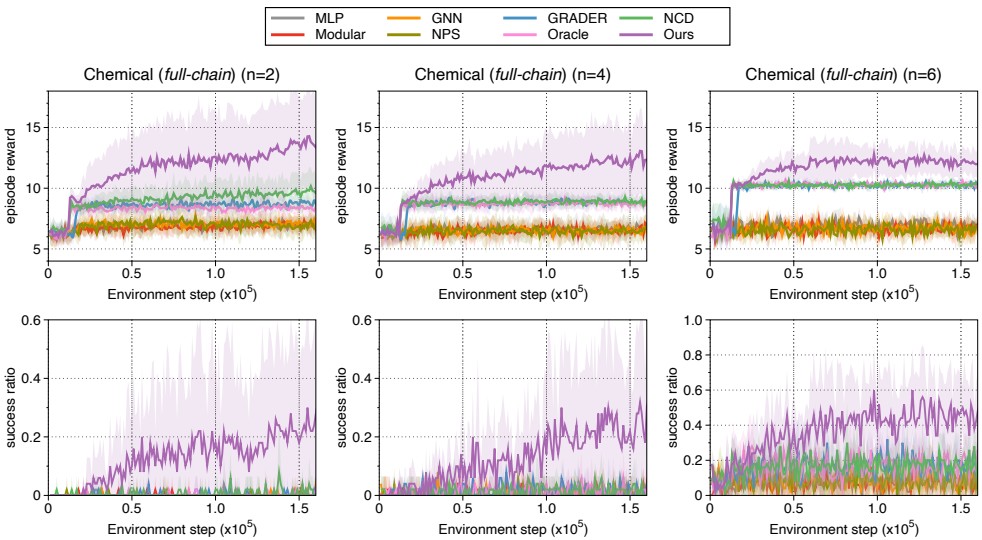

Figure 20: Learning curves on downstream tasks in Chemical (*full-chain*) as measured on the episode reward (**top**) and success rate (**bottom**).

