# OpenReview forum: "Quantized Local Independence Discovery for Fine-Grained Causal Dynamics Learning in Reinforcement Learning"
_ICLR.cc/2024/Conference — Submitted to ICLR 2024_

### Official Review · Reviewer_Ufsn · 2023-10-30

**Soundness:** 3 good
**Presentation:** 3 good
**Contribution:** 3 good
**Rating:** 6
**Confidence:** 3

**Summary:**

This paper proposed a novel method that discovers meaningful events and infers fine-grained causal relationships. They recommend learning a discrete latent variable representing the pair of events and causal graphs via vector quantization. Experimental results demonstrate their method is more robust to unseen states.

**Strengths:**

1.	The discovery of fine-grained event-based causality is novel and intuitive, and the experimental results also prove the author's point of view.
2.	The authors theoretically prove that their grouping method does not depend on the hyperparameter k setting and that meaningful context can be discovered in each group.
3.	Figures 4 and 5 in the experimental results clearly demonstrate the advantages of their event-based method in discovering fine-grained causal relationships.

**Weaknesses:**

1.	The validation environment for the experiment is somewhat simple, and the assumption of full observables seems to have significant limitations.

**Questions:**

1.	In Table 3, it seems that the larger K is, the better the performance is. What happens if K is increased to equal the number of samples?
2.	Intuitively, an event should be a sequence of continuous states and actions. I would like to ask whether Eq. (7) is more inclined to aggregate neighboring states?

---

> ### Author Response · Authors · 2023-11-19
>
> We thank the reviewer for the positive assessment and insightful remarks. We respond to each of your comments below:
>
> ---
>
> > **[W1]** The validation environment for the experiment is somewhat simple, and the assumption of full observables seems to have significant limitations.
> >
> - Discovering fine-grained causal relationships (i.e., local causality) from the high-dimensional image observation is certainly an important future direction. However, it has not been explored even in a fully-observable setting. We would like to note that our work is the first step towards discovering and utilizing fine-grained causal relationships for enhancing model-based RL. We indeed agree with the reviewer that it is a promising future direction to further investigate this in more complex environments and in a partially-observable setting.
>
> > **[Q1]** In Table 3, it seems that the larger K is, the better the performance is. What happens if K is increased to equal the number of samples?
> >
> - In our experiments, the total number of samples we used is 15k and 20k for Chemical and Magnetic, respectively. We speculate that as $K \rightarrow \infty$, it would suffer from overfitting similar to NCD, a sample-specific inference method.
>
> > **[Q2]** Intuitively, an event should be a sequence of continuous states and actions. I would like to ask whether Eq. (7) is more inclined to aggregate neighboring states?
> >
> - We speculate that Eq. 7 would be inclined to aggregate neighboring states because we use neural networks for an embedding function in practice.
>
> ---
>
> We again appreciate the reviewer for the valuable comments to improve our manuscript. We tried our best to address the reviewer’s concerns and questions, and we incorporated our responses into our revised manuscript. If our responses are insufficient or if there is a misunderstanding of the reviewer’s questions, we are happy to engage in further discussions.

---

### Official Review · Reviewer_eMfR · 2023-11-01

**Soundness:** 2 fair
**Presentation:** 3 good
**Contribution:** 3 good
**Rating:** 6
**Confidence:** 1

**Summary:**

This paper considers using the subgraph decomposition method under certain event conditions to enhance the generalization and robustness of the. The first step is to find the partition of events. Then, based on the partitions, a causal discovery method is applied to find the causal graph.

**Strengths:**

This paper presents an interesting idea, which suggests that causal relationships can be changed with variations in the conditional event. This is an issue worth discussing, especially in dynamic systems. The writing in this paper is easy to follow, and the method is supported by thorough experimental work.

**Weaknesses:**

See the questions below. If the authors can solve my concerns, I would like to raise my score.

**Questions:**

* How can we understand the concept of 'event'? Is it akin to a specific state value or an additional variable, similar to the domain ID in domain generalization tasks? Could you provide some examples in the context of reinforcement learning scenarios?
* Definition 1 may not fully describe an event-conditioned system. What if we consider a situation where changing the event alters the entire causal relationship? Definition 1 only discusses cases where the edges under certain conditions should be a subset of the original edge set.

* Can the score (equation 4) assist in identifying the causal graph? In Huang et al.'s paper, the score is a direct measure of conditional independence. Does equation 4 also measure conditional independence? In Brouillard et al.'s work, I found that the intervention data are required to support identifiability. Previous works have used score-based methods to discover causal graphs only under certain assumptions (e.g., linear and non-Gaussian Structural Causal Models). However, this paper lacks rigorous assumptions that substantiate identifiability.

* In equation 5, both h and e_j are learned. How can we ensure a nontrivial result? Specifically, if both h and e_j are learned as the static value 0, how can we avoid trivial outcomes?

* Although there are some positive results in Table 3 shows that considering the subgraph can help to enhance the causal discovery score. I’m still curious about why breaking the full graph into subgraphs by removing unnecessary edges can help enhance the robustness. If the edges are unnecessary, the causal function on the full graph will not consider the causal effect of it.

---

> ### Author Response · Authors · 2023-11-19
>
> We sincerely appreciate the reviewer’s time and efforts to provide constructive feedback to improve our paper. We respond to each of your comments below.
>
> ---
>
> > **[Q1]** How can we understand the concept of 'event'? Is it akin to a specific state value or an additional variable, similar to the domain ID in domain generalization tasks? Could you provide some examples in the context of reinforcement learning scenarios?
> >
> - Consider the mobile home robot. In general, it could navigate the rooms and interact with various objects. However, when the door is closed, it cannot interact with the objects in the other rooms; to do so, it would have to first opens the door. In the context of RL, the state variables (entities) consist of robot, door, and other objects. The specific event we are concerned with is the context of the door opened.
>
> > **[Q2]** Definition 1 may not fully describe an event-conditioned system. What if we consider a situation where changing the event alters the entire causal relationship? Definition 1 only discusses cases where the edges under certain conditions should be a subset of the original edge set.
> >
> - As the reviewer pointed out, the entire causal relationship could change in general (e.g., changes through time [1]). In our work, we consider the setting where the underlying causal relationships remain stationary. In this setting, LCGs are always subgraphs of the (global) causal graph because if a causal relationship (i.e., edge) is absent in the whole domain, then it is also absent in any arbitrary event.
>
> > **[Q3]** Can the score (equation 4) assist in identifying the causal graph? In Huang et al.'s paper, the score is a direct measure of conditional independence. Does equation 4 also measure conditional independence? In Brouillard et al.'s work, I found that the intervention data are required to support identifiability. Previous works have used score-based methods to discover causal graphs only under certain assumptions (e.g., linear and non-Gaussian Structural Causal Models). However, this paper lacks rigorous assumptions that substantiate identifiability.
> >
> - Sorry for inconvenience. Our score function is the regularized maximum likelihood score, similar to [2]. In contrast to [2], which used interventional data to identify the graph up to $\mathcal{I}$-MEC, we use observational data to identify the graph up to Markov equivalence class (MEC). In our setting of factored MDP, each MEC contains a unique graph because temporal precedence determines the orientation of the edges. We also assume that our model using a neural network has sufficient capacity to model the ground truth density (Assumption 3 in Appendix B in the revised manuscript). Throughout the Appendix B in our revised manuscript, we have included rigorous assumptions and proofs that substantiate identifiability.
>
> > **[Q4]** In equation 5, both h and e_j are learned. How can we ensure a nontrivial result? Specifically, if both h and e_j are learned as the static value 0, how can we avoid trivial outcomes?
> >
> - In theory, $\mathcal{L}_{\texttt{pred}}$ in eq. 6, i.e., regularized log likelihood loss, also updates $h$, and thus affects the learning of latent codes $e_j$. Thus this prevents the degenerate solution, e.g., constant static $h$, because such trivial solution that learns a single causal graph over the entire domain has higher regularization loss.
> - In practice, the training of vector quantization still often becomes unstable and yields a degenerate solution even though it achieves higher loss: a phenomenon called *codebook collapsing* in VQ-VAE literature. A popular technique to prevent such collapsing and stabilize the training is to use exponential moving average (EMA) to update the codebook, which is also adopted in our implementation. We discuss in Appendix C.3.3 recent relevant techniques and tricks to further stabilize the training.

---

> > ### Author Response · Authors · 2023-11-19
> >
> > > **[Q5]** Although there are some positive results in Table 3 shows that considering the subgraph can help to enhance the causal discovery score. I’m still curious about why breaking the full graph into subgraphs by removing unnecessary edges can help enhance the robustness. If the edges are unnecessary, the causal function on the full graph will not consider the causal effect of it.
> > >
> > - Consider the above example of mobile home robot. If the door is closed, the objects on the other side of the door become irrelevant, thus the robot should keep function well even if there are unseen objects on the other side of the door. In other words, these edges are *locally unnecessary* only under the specific events. They are relevant in general, thus (learned) causal function using the full graph would be affected by these edges. Consequently, this harms their prediction accuracy and RL performance as shown in Table 1-3.
> >
> > ---
> >
> > We again deeply thank the reviewer for the insightful and constructive comments to improve our manuscript. We tried our best to address the reviewer’s concerns and questions, and we incorporated our responses into our revised manuscript. If our responses are insufficient or if there is a misunderstanding of the reviewer’s questions, we are happy to engage in further discussions.
> >
> > **References**
> >
> > [1] Factored Adaptation for Non-stationary Reinforcement Learning, NeurIPS 2022
> >
> > [2] Differentiable Causal Discovery from Interventional Data, NeurIPS 2020

---

> ### Author Response · Authors · 2023-11-23
> **Gentle reminder.**
>
> Dear reviewer `eMfR`,
>
> As the discussion period is ending soon, we are wondering if our responses and revision have addressed your concerns. We find the constructive comments from the reviewer extremely helpful in improving our initial manuscript, and we would be happy to engage in further discussions if you have any additional questions or suggestions. If our responses have addressed your concerns, we would highly appreciate it if you could re-evaluate our work based on our responses and revised manuscript.
>
> Best regards,
>
> Authors of the submission 4522.

---

### Official Review · Reviewer_JfxM · 2023-11-01

**Soundness:** 2 fair
**Presentation:** 3 good
**Contribution:** 3 good
**Rating:** 6
**Confidence:** 3

**Summary:**

This paper proposes a novel reinforcement learning method by considering causal relationships. Specifically, the authors jointly optimize the l1 regularized likelihood and a vector quantization loss, during which the event and each event-specific causal graph are expected to be identified. Experimental results on two environments show the effectiveness of the proposed method.

**Strengths:**

1. The method looks novel to me.

2. Overall speaking the identifiability theory looks plausible to me.

3. The paper is well-written and easy to follow.

**Weaknesses:**

1. The quantization technique in eq 5 is not novel. VQVAE and related papers should be cited there.

2. Though overall the identifiability theory is plausible, however, the identifiability of events is unclear to me. See my questions below.

**Questions:**

1. Why an event-specific graph has to be constrained to be a subgraph of the underlying G?

2. Why vector quantization can correctly identify the event?

3. Does the identifiability in Theorem 1 rely on the specific choice of K?

---

> ### Author Response · Authors · 2023-11-19
>
> We thank the reviewer for the positive assessment and insightful remarks. We respond to each of your comments below:
>
> ---
>
> > **[W1]** Citation of VQ-VAE
> >
> - Indeed, we adapt the quantization technique of VQ-VAE, where we cited in eq. 7, and provide discussions regarding the recent techniques related to VQ-VAE in Appendix C.3.3. Following the reviewer’s suggestion, we have also added citation in eq. 5 in the revised manuscript. As a side note, our utilization of vector quantization is quite different from most of the works on learning discrete latent codebook with vector quantization whose main focus is the reconstruction of the observation.
>
> > **[Q1]** Why an event-specific graph has to be constrained to be a subgraph of the underlying G?
> >
> - In certain events, causal relationships may no longer exist. LCGs are always subgraphs of the (global) causal graph because if a causal relationship (i.e., edge) is absent in the whole domain, then it is also absent in any arbitrary event.
>
> > **[Q2]** Why vector quantization can correctly identify the event?
> >
> - We use vector quantization to partition the state-action space. By imposing the dynamics model to use sparse subgraphs for each corresponding partition, vector quantization tends to cluster samples that causal relationships are explained by the same LCG.
>
> > **[Q3]** Does the identifiability in Theorem 1 rely on the specific choice of K?
> >
> - Thm. 1 holds for any choice of $K\geq 2$. If $K=1$, it is indeed impossible to identify particular event $\mathcal{E}\subsetneq \mathcal{X}$ which induces fine-grained causal relationships.
>
> ---
>
> We again appreciate the reviewer for the valuable comments to improve our manuscript. We tried our best to address the reviewer’s concerns and questions, and we incorporated our responses into our revised manuscript. If our responses are insufficient or if there is a misunderstanding of the reviewer’s questions, we are happy to engage in further discussions.

---

> ### Comment · Reviewer_JfxM · 2023-11-22
>
> Thank you for your response. After reading all the review comments I tend to keep my rating unchanged.

---

### Official Review · Reviewer_RHPG · 2023-11-01

**Soundness:** 1 poor
**Presentation:** 1 poor
**Contribution:** 2 fair
**Rating:** 5
**Confidence:** 3

**Summary:**

The paper proposes a gradient-based method for local-independence-based causal discovery, and applies it to the Causal Dynamics Modeling problem in model-based reinforcement learning. Following (Hwang et al. 2023), the basic idea is to partition the state-action space into subsets, each called an "event" in this paper, and then to learn the transition function of the MDP as a collection of event-specific causal models. The main difference from prior work is that the event partitioning in this paper is learned through a clustering procedure over a (jointly-learned) latent embedding of the state-action space, using Eq.7. With two specially-designed MDP environments, the paper experimentally shows that MDP planning powered by dynamics model learned from their method performs more robustly under some out-of-distribution settings. Finally, the paper also presents some theoretical results about the soundness of jointly optimizing the event partitioning scheme and the event-specific causal models.

**Strengths:**

I think the paper touches an interesting and important topic. Causal discovery, for whatever reason we want to do it, is often hindered by the fact that many causal effects only manifest themselves under certain conditions. So, targeting at learning local models that each only works for a specific condition, instead of at learning a generic model, seems to be a promising direction for causal discovery in general. The problem of collaboratively learning the condition-specific models and the conditions that effectively facilitate such model discovery, is however quite challenging, and is attacked in this paper. It is also nice to see that the paper embeds this causal discovery task into model-based RL and experimentally tests its application in this scenario.

**Weaknesses:**

**(a)** I am not sure about the soundness of the proposed method. The paper claims that they are jointly optimizing both the event decomposition/partitioning scheme and the causal models, but I doubt if the learning of the decomposition scheme is indeed effectively signaled by the impact of the decomposition on the quality of the resulted causal models. Moreover, I doubt if the learning objective designed for optimizing decomposition (Eq.7) may lead to degenerate results. See my question 1 and 2 below for the detailed concerns. Since joint optimization is a main selling point of this paper (without it the theory part becomes largely irrelevant to the experiment part, for example), this is a serious issue that must be clarified.

**(b)** I have concern on the reproducibility of the experiment part. No pseudo-code is provided for the proposed method, and the method description in Section 3.3, 4.1.2, and C.2, missed some important pieces. For example, how is the data sampled (what's the exploration policy being used)? Exactly what is being updated by L_pred and L_quant (the gradient is respective to what)? What are exactly computed for the differentiation?

Moreover, the experimentation code is not given either, and the description of the two MDP environments are not detailed enough for third-parties to reproduce the experiment, I'm afraid. For example, how exactly the nodes in the Chemical environment affect each other? How exactly the noise is injected at testing time? In the robot-arm environment, how exactly the "unseen locations" of the box is determined at testing time? In Section C.1.3, it's said that at training time the positions of the box are "randomly sampled within the range of the table", does this mean every location on the table has a positive probability density to be selected at training time (If so, the location of the box at testing time is not really out-of-distribution, even though it may be unseen)?

**(c)** I have concerns on the theory part too. On one hand, the main theorem seems to only apply to a rather special case (where the causal model can only have two irreducible phases). On the other hand, the lemmas (Prop. 1-3) used for proving the main theorems seem to have flawed statements and proofs. Some important assumptions are not explicitly stated. See my question 3~8 for the details in this regard.

**(d)** The paper may need to better contrast to prior work, especially to (Hwang et al., 2023). It seems (Hwang et al., 2023) is not limited to sample-specific decomposition, but also discusses event-level or event-set-level decomposition. Proposition 1 in the paper under review is essentially a rephrasing of Proposition 4 in (Hwang et al., 2023), for example.

---  post-rebuttal edits ---

After the rebuttal I have less concerns in all the four aspects above, but not to the extent that I'm confident to say that the issues have been resolved. Overall, I raise my recommendation to 5 (marginally reject); I think it may be beneficial to have another round of revision and review for this paper, although I won't strongly object to accepting the current draft right away.

**Questions:**

1. In Eq.6, will the gradient of L_pred be used to update the codebook $C$ and (parameters in) the embedding model $h$? If I understand correctly, the decomposition scheme mainly depends on the codebook $C$ and the embedding model $h$, and both of them affect L_pred through $e$, via Eq.5 (and $e$ affects L_pred further through $A$). However, how does the gradient propagates through the argmin operator in Eq.5?

(Suppose the gradient of L_pred does not change $C$ and $h$, then the learning of the decomposition scheme actually does not take into account its consequence on the likelihood score, but is solely based on the gradient of L_quant, in this case the algorithm you proposed is not truly a joint optimization method, although the decomposition and the causal models are indeed learned "in parallel".)

2. I don't quite get the rationale behind Eq.7. It seems to me that the L_quant defined in Eq.7 wants to encourage the feature vector $h$ of the state-action and its corresponding cluster center $e$ to be close to each other. But is this enough to learn a "good" decomposition? As a trivial and degenerate solution, imagine the embedding model h always outputs a constant vector, and all the cluster center $e_z$ in the codebook also equal this constant vector, this will minimize L_quant to zero, but is clearly not what we want. This question is assuming that the answer to Question 1 above is no (that the learning of $C$ and $h$ are not further based on the likelihood loss L_pred).

3. I can't quite follow the proof of Proposition 2. In fact, I'm not sure if the proposition is exactly correct as there is no requirement on the value of lambda at all. In comparison, Lemma 1 requires *small enough* (yet non-zero) lambda. Does Prop.2 also require special lambda values? For continuous state and action spaces, there can be an uncountably infinite number of possible decomposition schemes, each may require a different "small enough" lambda for the regularized likelihood score to work, in this case do we still have a single lambda that works for all decompositions? In general, the role of lambda seems to be subtle yet crucial for the entire theory developed here.

4. In the proof of Proposition 3, I'm not sure about the inequality at line 4 of the proof -- between the equality sign, why can the likelihood terms $\hat{p}$ in S be dropped?

5. The probability factorization at Line 1 of Page 3 implies that you are assuming all the $S'_j$'s are independent to each other (conditioned on given $s,a$), right? Which propositions proved in the paper need this assumption? Without this assumption, the causal model is not necessarily bipartite, and skeleton learning based on local independence would be not enough, and we perhaps don't have unique identifiability any more? Also, does the environments used in your experiment satisfy this assumption?

6. Please explicitly and formally state all the external propositions that your proofs crucially depend on and yet are only proved in other literature, such as (Hwang et al. 2023, Prop. 4) and (Brouillard et al. 2020, Thm. 1).

7. You define "local causal model" based on the concept of PA(j; E), but how is PA(j;E) defined? Does the global causal model entail the local models through the induced local independence relationships? In that case a proper *definition* of local causal model would be entirely conditioned on the E-faithfulness assumption, am I right?

8. In the proof of Lemma 1 you mentioned some assumptions and said they are assumed "throughout the paper". Such assumptions should not be placed in the appendix, in the middle of a proof for a proposition. They should at least be given in the statement of the involved propositions, in the main text.

9. Is your algorithm equivalent to NCD if we set $K=|S \times A|$?

---

> ### Author Response · Authors · 2023-11-19
>
> We sincerely appreciate the reviewer’s time and efforts to provide constructive feedback to improve our paper. We respond to each of your comments below.
>
> ---
>
> ### Questions for the proposed method
>
> > **[Q1-2]** Backpropagation in Eq. 5-7. [..] Exactly what is being updated by $\mathcal{L}\_{\texttt{pred}}$ and $\mathcal{L}_{\texttt{quant}}$? What are exactly computed for the differentiation?
> >
> - Gradient of $\mathcal{L}_{\texttt{pred}}$ (eq. 6) **does** update the latent embedding $h$, and consequently, affects the codebook $C$.
>     - First, recall that $A\sim g_{\texttt{dec}}(e)$, backpropagation from $A$ in $\mathcal{L}\_{\texttt{pred}}$ updates the quantization decoder $g_{\texttt{dec}}$through $e$.
>     - Second, we copy gradients from $e$ (= input of $g_{\texttt{dec}}$) to $h$ (= output of $g_{\texttt{enc}}$) during the backward path in eq. 5, following a popular trick used in VQ-VAE [1]. By doing so, $\mathcal{L}\_{\texttt{pred}}$ updates the quantization encoder $g_{\texttt{enc}}$ and $h$.
>     - Finally, this also affects the learning of the codebook $C$ since $C$ is updated with $\mathcal{L}_{\texttt{quant}}$ (eq. 7) which involves in $h$.
> - To sum up, $\mathcal{L}\_{\texttt{pred}}$ updates the encoder $g_{\texttt{enc}}(s,a)$, decoder $g_{\texttt{dec}}(e)$, and the dynamics model $\hat{p}$. $\mathcal{L}\_{\texttt{quant}}$ updates $g_{\texttt{enc}}$ and the codebook $C$.
> - Since the learning of $C$ and $h$ are also based on $\mathcal{L}_{\texttt{pred}}$, this prevents yielding a degenerate solution such as constant vector $h$.
> - We have included our responses and further details on our method and its implementations in our revised manuscript (Appendix C.2.1). We promise to publicly open-source the experimentation code after the paper being published.
>
> ---
>
> ### Questions for the experiment
>
> > How is the data sampled?
> >
> - We use a random policy for the initial data collection.
>
> > Detailed description of two environments.
> >
> - In Chemical, the underlying color-change mechanism of each node given its parents follows a predefined conditional probability table implemented with a randomly initialized neural network. To create OOD states, we use a noise sampled from $\mathcal{N}(0, \sigma^2)$, similar to [2]. In Chemical, the noise is multiplied to the one-hot encoding representing color during the test.
> - In Magnetic, the box position is sampled from $\mathcal{N}(0, \sigma^2)$ during the test. We note that the box can be located outside of the table, which never happens during training. We use $\sigma=100$ for both environments.
>
> ---
>
> ### Questions for the theory
>
> > **[Q3]** Requirement on the value of $\lambda$.
> >
> - Prop. 2 holds for small enough $\lambda$, i.e., $0<\forall\lambda\leq \eta(\\{\mathcal{E}\_z\\})$, where we have included detailed derivations in the proof of Prop. 2 in our revised manuscript. We also note that $\eta(\\{\mathcal{E}\_z\\})>0$ is a value specific to a decomposition. Thus, for Prop. 3, we introduce Assumption 5 (on page 18) which states that there exist a lambda that works for all decompositions, i.e., $\inf_{\\{\mathcal{E}\_z\\}\in \mathcal{T}} \eta(\\{\mathcal{E}\_z\\}) > 0$ where $\mathcal{T}$ is a set of all decompositions. We also discuss the violation of this assumption: in that case, the arguments hold for decompositions $\mathcal{T}\_\lambda=\\{\\{\mathcal{E}\_z\\}\mid \eta(\\{\mathcal{E}\_z\\})\geq \lambda \\}$, where $\mathcal{T}\_\lambda \rightarrow \mathcal{T}$ as $\lambda \rightarrow 0$ (see page 18 in our revised manuscript).
>
> > **[Q4]** The last equality in the proof of Prop. 3.
> >
> - We have also included more detailed derivations for the proof of Prop. 3 in our revised manuscript. In short, for the arbitrary decomposition $\\{\mathcal{E}\_z\\}$ and its corresponding *true* LCGs $\\{\mathcal{G}\_z\\}$, $\mathcal{S}(\\{\mathcal{G}_z,\mathcal{E}_z\\}\_{z=1}^K)=\mathbb{E}\_{p(s, a, s')} \log p(s'\mid s, a) - \lambda \sum_z p(\mathcal{E}_z) \cdot \lvert \mathcal{G}_z \rvert$, and thus the first term is canceled out in the last equality in the proof of Prop. 3.

---

> > ### Author Response · Authors · 2023-11-19
> >
> > > **[Q5]** Factored MDP setting.
> > >
> > - The probability factorization $p(s'\mid s, a) = \prod_j p(s'_j\mid s, a)$ implies that each $S'_j$ conditioned on $S, A$ are independent to each other. This is the setting of factored MDP where our paper and this line of related works commonly considers. Our propositions and theorem are developed under this setting since we assume a bipartite causal graph. As the reviewer pointed out, a causal graph is not necessarily bipartite without this assumption, and we would no longer have unique identifiability anymore. In our experiments, Magnetic satisfies the factored MDP assumption but not Chemical, which contains instantaneous effects.
> >
> > > **[Q6]** Please explicitly and formally state all the external propositions
> > >
> > - Sorry for the inconvenience. We have made the external proposition explicit and improved the clarity by adding detailed derivations throughout Appendix B in our revised manuscript.
> >
> > > **[Q7]** how is $Pa(j; \mathcal{E})$ defined? Does the global causal model entail the local models through the induced local independence relationships? In that case a proper *definition* of local causal model would be entirely conditioned on the E-faithfulness assumption, am I right?
> > >
> > - We have added formal definition of $Pa(j; \mathcal{E})$ in Def. 5 on page 15 in our revised manuscript. In words, $Pa(j; \mathcal{E})$ is a minimal subset of $Pa(j)$ in which the local independence on $\mathcal{E}$ holds. And Yes, the global causal model entails the local models through the induced local independence relationships. To clarify, LCG always exists because $Pa(j; \mathcal{E})$ always exists regardless of the $\mathcal{E}$-faithfulness assumption. $\mathcal{E}$-faithfulness implies that $Pa(j; \mathcal{E})$ is unique, and thus implies the uniqueness of LCG $\mathcal{G}_\mathcal{E}$.
> >
> > > **[Q8]** Assumptions.
> > >
> > - Sorry for the inconvenience. Following the reviewer’s suggestion, we have made our assumptions explicit, and they are now given in the statement of the involved propositions. Due to the space constraints, precise statement of the assumptions are currently presented before the proof, in the Appendix B in our manuscript.
> >
> > ---
> >
> > ### Questions for the prior work
> >
> > > The paper may need to better contrast to prior work [..] It seems (Hwang et al., 2023) is not limited to sample-specific decomposition, but also discusses event-level or event-set-level decomposition.
> > >
> > - To clarify, Hwang et al., (2023) indeed discussed event-level decomposition, which our work draws inspiration from. However, their proposed method does not explicitly discovers such decomposition, but only infers LCG for each sample. In contrast, our method explicitly discovers $\\{\mathcal{G}_z, \mathcal{E}_z\\}$, decomposition and corresponding LCGs. We have included related discussions in Appendix C.3.3 in our revised manuscript to better contrast to prior work.
> >
> > > **[Q9]** Is your algorithm equivalent to NCD if we set $K=|\mathcal{S}\times\mathcal{A}|$ ?
> > >
> > - Correct. We have also included this relationship in Appendix C.3.3 in the revised manuscript.
> >
> > ---
> >
> > We again deeply appreciate the reviewer for the constructive comments to improve our manuscript. We find them extremely helpful to improve the clarity and polish our initial manuscript. We tried our best to address the reviewer’s concerns and questions, and we incorporated our responses into our revised manuscript. If our responses are insufficient or if there is a misunderstanding of the reviewer’s questions, we are happy to engage in further discussions.
> >
> > **References**
> >
> > [1] Neural Discrete Representation Learning, NIPS 2017
> >
> > [2] Causal Dynamics Learning for Task-Independent State Abstraction, ICML 2022

---

> > ### Comment · Reviewer_RHPG · 2023-11-23
> > **regarding the rationale of the proposed method (Q 1-2)**
> >
> > Thanks for the elaborations and for the paper revisions.
> >
> > > we copy gradients from $e$ (= input of $g_{\texttt{dec}}$) to $h$
> >
> > First, this is an important detail and an essential step in your method that needs to be clearly stated in the main text, such as in Section 3.3; mentioning it in appendix is not enough, in my opinion.
> >
> > With this additional step in mind, I am still not really sure about the soundness of the proposed method. According to Eq.(5), $e$ is the cluster center of the cluster that $h$ is assigned to based on L_2 norm. Consider a small change of $h$. If this small change on $h$ causes a re-assignment of the cluster (i.e. causes a change of $z$), then $e$ changes completely to another cluster center. If there is no re-assignment of the cluster, then the small change on $h$ has no impact on L_pred thus the gradient is 0. The above reflection suggests that $L_pred$ does not have well-defined gradient with respect to $h$ at all.
> >
> > Now you propose to continue the gradient propagation despite the fact that $e \neq h$ (they are not necessarily close to each other even) and that the gradient $\nabla_h L_{pred}$ does not really exist. Operationally this is done by the gradient-copy operation as you mentioned, and you justify this trick using literature about VQ-VAE. The explanation of the rationale is thus not self-contained from here, and thus my concern about the theoretical soundness of your method remains.
> >
> > As a bottom line, I suggest you to explain this trick, including how it is justified in another literature, in the Preliminary section of your paper.

---

> > ### Comment · Reviewer_RHPG · 2023-11-23
> > **regarding the theory part (Q 3-8)**
> >
> > It appears that you have made major revisions in this part, including re-organized assumptions (with some of them newly introduced as response to my question), re-stated propositions, and re-written proofs. While I do appreciate this effort, I am sorry that I don't have the bandwidth at this moment to verify the revised theory. As reviewer I already carefully read through the main text as well as most appendix sections of the originally submitted paper; verifying the revision at this level requires me to dive into all the math details again.
> >
> > As a general and *minor* feedback to your rebuttal here: In my humble opinion, I think clearly stating all assumptions and all external propositions that your theory relies on are not something about "convenience" as you might be hinting, but really about the soundness of the theory you developed. If the technical soundness of the theory is not really important for the paper, I suggest to just move the entire "theory part" into appendix, and leave the main text focus on clearly explaining your proposed method and your experiments.

---

> ### Author Response · Authors · 2023-11-23
> **Gentle reminder.**
>
> Dear reviewer `RHPG`,
>
> As the discussion period is ending soon, we are wondering if our responses and revision have addressed your concerns. We find the constructive comments from the reviewer extremely helpful in improving our initial manuscript, and we would be happy to engage in further discussions if you have any additional questions or suggestions. If our responses have addressed your concerns, we would highly appreciate it if you could re-evaluate our work based on our responses and revised manuscript.
>
> Best regards,
>
> Authors of the submission 4522.

---

> > ### Comment · Reviewer_RHPG · 2023-11-23
> > **regarding contrast to (Hwang et al., 2023) (Q 9)**
> >
> > You claim that (Hwang et al., 2023) focuses on "*infers LCG for each sample*". But it seems to me that the concept of "context-set" in that paper corresponds exactly to the concept of "event" in your paper, and that paper seems to indeed study the problem of learning local causal model for a context-set (which is not necessarily a single context)? Such context-set decomposition appears to be also learned in that paper, as evidenced by the title of its section 4 "*Discovering Local Independence Relationships by Learning the Partition*". Am I misunderstanding something in that paper here?

---

> > > ### Author Response · Authors · 2023-11-23
> > >
> > > ### Response to “regarding the rationale of the proposed method (Q 1-2)”
> > >
> > > - Following the reviewer’s suggestion, we will include background on VQ-VAE and training details in the Preliminary section of the final version.
> > >
> > > > The above reflection suggests that $\mathcal{L}_{\texttt{pred}}$ does not have well-defined gradient with respect to $h$ at all.
> > > >
> > >
> > > $$
> > > e = e_z, \quad \text{where} \quad  z = \text{argmin}\_{j \in \left[K\right]} \\|h - e_j\\|_2.  \quad \text{(Eq.5)}
> > > $$
> > >
> > > - While there is no real gradient defined for $h=g_{\texttt{enc}}(s, a)$ in Eq. 5, we approximate the gradient using the trick of VQ-VAE, which copies the gradient $\nabla_e \mathcal{L}\_{\texttt{pred}}$ to update the encoder $g_{\texttt{enc}}$, similar to straight-through estimator.
> > > - The rationale behind this trick is that the gradients contain useful information for how the encoder $g_{\texttt{enc}}$ has to change its output $h=g_{\texttt{enc}}(s, a)$ to lower the prediction loss $\mathcal{L}_{\texttt{pred}}$.
> > >     - This gradient $\nabla_e \mathcal{L}_{\texttt{pred}}$ will update the encoder to change its output $h$, which could alter the quantization (i.e., assignment of the cluster) in the next forward pass.
> > >     - A larger prediction loss (which implies that this sample $(s, a)$ is assigned to the wrong cluster) induces a bigger change on $h$ and consequently it would be more likely to cause a re-assignment of the cluster.
> > >
> > > ### Response to “regarding the theory part (Q 3-8)”
> > >
> > > - We regret using the term "inconvenience" in our earlier messages. We hold the view that the technical soundness of the theory is important and, thus, fully agree that clearly stating all external propositions and assumptions is crucial (currently, they are presented in the Appendix; however, they will be integrated into the main text as needed). We again deeply appreciate the reviewer’s feedback, which made us improve the rigor and soundness that is incorporated in the revised manuscript.
> > >
> > > ### Response to ''regarding contrast to (Hwang et al., 2023) (Q 9)''
> > >
> > > - To clarify, such decomposition is indeed first studied in the prior work (Hwang et al., 2023), but their method (i) lacks theoretical guarantees to discover true LCGs, and (ii) does not **explicitly** identify events.
> > >     - The method introduced in (Hwang et al., 2023) does not *explicitly* identify events since it only learns the function that maps each sample $(s, a)$ to LCG (i.e., sample-specific inference). Here, the events are *implicitly* identified: the samples having the same inferred LCG belong to the same event. Due to its limitation, important attributes of the decomposition such as the number of discovered events (i.e., size of the decomposition) are unclear, and it lacks theoretical guarantee.
> > >     - In contrast, our method learns the function that maps each code $e$ to LCG (i.e., event-specific inference) and events are *explicitly* identified by the latent clustering. Thus, the identified events are more interpretable and our method is also theoretically-grounded.

---

### Official Review · Reviewer_s5uf · 2023-11-01

**Soundness:** 3 good
**Presentation:** 3 good
**Contribution:** 3 good
**Rating:** 6
**Confidence:** 3

**Summary:**

This paper presented a local causal dynamic learning method by introducing meaningful events. It consists of two parts (1) inferencing local causal graph through quantization and (2) predicting next state. Empirical result on two RL environments showed the effectiveness of the proposed learning method. More importantly it also showed identifiability result for the proposed method.

**Strengths:**

1. The assumption that separating the whole (s,a) space into decomposition is intuitive.
2. This paper provided identifiability result of the proposed method that the optimal decomposition that maximizes the score identifies a meaningful context that exhibits fine-grained causal relationships.
3. The experiments (Chemical and Magnetic) showed the effectiveness of the proposed method.

**Weaknesses:**

* The theory and the implementation seem not aligned.
  * In theory there are ground truth decomposition which the LCG are the same within partition and different otherwise, however, it seems that for Magnetic, there is a uniform causal graph which does not change locally.
  * Identifiability results are not reflected in the method. There is no quantitative result that supporting that identifiability is or is not achieved in the experiment.
  * Unclear relation between ID, OOD setting with the decomposition and the codebook size K (see questions).
* The assumption on the decomposition is not mild. In real world data, the transition function is usually highly complex. This method can only handle the case when different parts in the decomposition have different LCG. However, in real case, all parts may share the same LCG due to the highly complex transition function. Even though the conditional independence relations are the same (same LCG), the transition function itself may be used to further partition the sample space into further small parts. That is saying same LCG but with different $p(s'\vert s,a)$.

**Questions:**

* Minor presentation issues about the notations.
  * In Sec 3.1, what is the definition of event $\mathcal{E}$?
  * In terms of the score function $\mathcal{S}(\\{\mathcal{G}_z,\mathcal{E}_z\\}_1^K):=\text{sup }\mathbb{E}\left[\text{log}\hat{p}(s' \vert s,a;\mathcal{G}_z) - \lambda \vert \mathcal{G}_z\vert\right]$
    * Which variables are for sup?
    * Which variables are for $\mathbb{E}$?
* In Proposition 3. why $\mathbb{E}\left[\vert\mathcal{G}_z\vert\right] \le \mathbb{E}\left[\vert\mathcal{\hat{G}}_z\vert\right]$.
  * Specifically, in appendix B.2 proof of Proposition 3 why the last equality holds? Can you provide detailed derivation involving the definition of score function $\mathcal{S}(\cdot)$?
* In the theory there is a ground truth decomposition $\\{\mathcal{G}_z,\mathcal{E}_z\\}_1^K$. How can this decomposition be aligned with the experiments? i.e., what are the ground truth decomposition and the corresponding LCGs in both Chemical and Magnetic environment?
* How to measure if the identifiability is achieved during learning process? Some measure of the distance between the learned LCG and the true LCG may be used here.
* For ID setting, the proposed method is not the best, is there any explanation for this?
* How to decide the number of codebook vector K?
* For the case mentioned in Weakness, when dealing with more complex data, if the LCG is shared but the transition function is indeed different among partitions, is the proposed method flexible enough to generalize to those case?

---

> ### Author Response · Authors · 2023-11-19
>
> We sincerely appreciate for reviewing our paper and providing us with constructive feedback. We respond to each of your comments below:
>
> ---
>
> > it seems that for Magnetic, there is a uniform causal graph which does not change locally. [..] what are the ground truth decomposition and the corresponding LCGs in both Chemical and Magnetic environment?
> >
> - In Magnetic, the context $D=$ {at least one of the objects is non-magnetic, i.e., colored black} induces the fine-grained causal relationships, and its corresponding LCG is shown in Fig. 7(c). The (global) CG is shown in Fig. 7(a) in our revised manuscript.
> - In Chemical, the context $\mathcal{D}=$ {color of the root node is red or blue} induces the fine-grained causal relationships and its corresponding LCG is shown in Fig. 4(c). The (global) CG is the lower triangular matrix.
>
> > There is no quantitative result that supporting that identifiability is or is not achieved in the experiment. [..] How to measure if the identifiability is achieved during learning process? Some measure of the distance between the learned LCG and the true LCG may be used here.
> >
> - The evaluation of local causal discovery is provided in Appendix C.3.2 and Fig. 16. We use the structural hamming distance (SHD), a metric used to quantify the dissimilarity between two graphs (i.e., the distance between the learned LCG and true LCG). We also provide an analysis of whether our method successfully identifies meaningful events through the codebook histograms and learned LCGs (e.g., Fig. 9).
>
> > Minor presentation issues about the notations. […] In Proposition 3. why $\mathbb{E}\left[\lvert\mathcal{G}_z\rvert\right] \leq \mathbb{E}\left[\lvert \hat{\mathcal{G}}_z\rvert \right]$? Can you provide detailed derivation?
> >
> - Sorry for the inconvenience. We have uploaded the revised manuscript to improve the clarity. We have included a precise statement of the score function and derivation in Eqs. 8-12 in Appendix B in our revised manuscript. Specifically,
>     - Event $\mathcal{E}$ is a subset of the joint space action space, i.e., $\mathcal{E}\subseteq \mathcal{X}= \mathcal{S}\times \mathcal{A}$.
>     - Score function is $\mathcal{S}(\\{\mathcal{G}_z,\mathcal{E}_z\\}\_{z=1}^K):=\text{sup}\mathbb{E}\left[\text{log}\hat{p}(s' \vert s,a;\\{\mathcal{G}_z, \mathcal{E}_z \\}) - \lambda \vert \mathcal{G}_z\vert\right]:=\text{sup}\_\phi \mathbb{E}\_{p(s, a, s')}\left[\text{log}\hat{p}(s' \vert s,a;\\{\mathcal{G}_z, \mathcal{E}_z \\}, \phi) - \lambda \vert \mathcal{G}_z\vert\right]$. (Eq. 8-12)
> - We have also included more detailed proof of Prop. 3 with derivations in our revised manuscript.
>     - In short, for an arbitrary decomposition $\\{\mathcal{E}_z\\}$ and its corresponding true LCGs $\\{\mathcal{G}_z\\}$, $\mathcal{S}(\\{\mathcal{G}_z,\mathcal{E}_z\\}\_{z=1}^K)=\mathbb{E}\_{p(s, a, s')}  \log p(s'\mid s, a) - \lambda \sum_z p(\mathcal{E}_z) \cdot \lvert \mathcal{G}_z \rvert$, and thus the first term is canceled out in the last equality in the proof of Prop. 3.
>
> > This method can only handle the case when different parts in the decomposition have different LCG. [..] Even though the conditional independence relations are the same (same LCG), the transition function itself may be used to further partition the sample space into further small parts. That is saying same LCG but with different $p(s'\mid s, a)$. […] is the proposed method flexible enough to generalize to those case?
> >
> - This is a great question! Recall eq. 3 that $p(s'_j\mid s, a)=p(s'_j\mid Pa(j; \mathcal{E}_z), z)$ for $(s, a)\in \mathcal{E}_z$, the dynamics prediction model takes not only $Pa(j; \mathcal{E}_z)$, but also $z$ as an input. This is because the transition function could be indeed different among parts with the same LCG, as the reviewer pointed out. Here, $z$ guides the network to learn (possibly) different transition functions even if the LCG is the same. Recall that each latent code $e_z\in C=\\{e_z\\}^K\_{z=1}$ denotes the partition, our model takes a one-hot encoding of size $K$ according to the latent code as the additional input to deal with such cases.
>
> > For ID setting, the proposed method is not the best, is there any explanation for this?
> >
> - During training (i.e., ID setting), all methods achieve similar near-optimal performances as shown in the learning curve (Fig. 18 on page 27).
>
> > How to decide the number of codebook vector K?
> >
> - In theory, the identifiability result holds for any choice of $K\geq 2$. In practice, we find that any choice of $K$ works reasonably well, except for $K=2$ where the training was often unstable (see relevant discussions in Appendix C.3.3).
>
> ---
>
> We again deeply thank the reviewer for the constructive comments. We tried our best to address the reviewer’s concerns and questions, and we incorporated our responses into our revised manuscript. If our responses are insufficient or if there is a misunderstanding of the reviewer’s questions, we are happy to engage in further discussions.

---

> > ### Comment · Reviewer_s5uf · 2023-11-23
> >
> > I appreciate the author's rebuttal and their response to my question. I have increased my score.

---

### Author Response · Authors · 2023-11-19
**General response to all reviewers.**

We deeply thank all reviewers for their invaluable feedback. We have individually addressed each reviewer's comments. We also uploaded a revised manuscript incorporating the following changes (highlighted in blue) with the main goal of improving clarity and rigor:

- We have explicitly and formally included detailed derivations, assumptions, and proof techniques that we originally borrowed from prior works (Appendix B in the revised manuscript).
- We have added detailed descriptions of the environments, implementation of our method, and additional discussions (Appendix C in the revised manuscript).

---

> ### Author Response · Authors · 2023-11-22
> **Gentle reminder.**
>
> Dear reviewers,
>
> Thank you again for your detailed feedback! As the discussion period will close in a day, we are wondering if you have any other questions. We are happy to engage in further discussions and address any remaining questions or concerns.
>
> Authors of the submission 4522.

---

### Meta-Review · Area_Chair_VLm2 · 2023-12-07

**Metareview:**

This paper introduces an innovative method for local causal dynamic learning in reinforcement learning. It utilizes quantization to infer local causal graphs and predict the next state. The approach is tested in two reinforcement learning environments, demonstrating its effectiveness. Notably, the paper also offers identifiability results for the proposed method, suggesting its potential to identify meaningful contexts with fine-grained causal relationships. On the other hand, the reviewers point out some weakness of the paper. For example, the theory and implementation appear misaligned, particularly in the Magnetic environment, where the causal graph does not seem to change locally, contradicting the theoretical framework. Moreover, the identifiability results, a key aspect of the paper, are not convincingly reflected in the method. There is an absence of quantitative results supporting whether identifiability is achieved in the experiments. Additionally, the relationship between in-distribution and out-of-distribution settings with the decomposition and the codebook size remains unclear. This ambiguity raises questions about the method's practical applicability in real-world scenarios, where transition functions are typically complex. At the discussion phase, the authors responded to these concerns by clarifying the context and corresponding Local Causal Graphs in the Magnetic and Chemical environments. They included an evaluation of local causal discovery in the appendix, using Structural Hamming Distance to quantify the dissimilarity between learned and true LCGs. Additionally, they revised the manuscript for improved clarity and addressed specific questions raised by the reviewers. However, despite these efforts, significant challenges remain. The theory appears to apply only to specific cases and is based on unexplained assumptions. There are also lingering doubts about the practical application of the proposed method in more complex environments and concerns regarding the reproducibility of the experiments. In light of these issues, the innovative aspects of the paper are overshadowed by the significant theoretical and practical challenges that remain unaddressed. The lack of clear alignment between theory and implementation, unresolved questions about identifiability and decomposition, and concerns about reproducibility are critical. These factors suggest that the paper, while ambitious and innovative in its approach to causal dynamics learning, is not yet ready for acceptance.

**Justification For Why Not Higher Score:**

The decision to not assign a higher score to the paper is primarily based on several unresolved issues, as described in the meta-review.

**Justification For Why Not Lower Score:**

N/A

---

### Decision · Program_Chairs · 2024-01-16

Reject